# How Ensembles of Distilled Policies Improve Generalisation in Reinforcement Learning

**Max Weltevrede**
Delft University of Technology
Delft, The Netherlands
`m.r.weltevrede@tudelft.nl`

**Moritz A. Zanger**
Delft University of Technology
Delft, The Netherlands
`m.a.zanger@tudelft.nl`

**Matthijs T. J. Spaan**
Delft University of Technology
Delft, The Netherlands
`m.t.j.spaan@tudelft.nl`

**Wendelin Böhmer**
Delft University of Technology
Delft, The Netherlands
`j.w.bohmer@tudelft.nl`

## Abstract

In the zero-shot policy transfer setting in reinforcement learning, the goal is to train an agent on a fixed set of training environments so that it can generalise to similar, but unseen, testing environments. Previous work has shown that policy distillation after training can sometimes produce a policy that outperforms the original in the testing environments. However, it is not yet entirely clear why that is, or what data should be used to distil the policy. In this paper, we prove, under certain assumptions, a generalisation bound for policy distillation after training. The theory provides two practical insights: for improved generalisation, you should 1) train an ensemble of distilled policies, and 2) distil it on as much data from the training environments as possible. We empirically verify that these insights hold in more general settings, when the assumptions required for the theory no longer hold. Finally, we demonstrate that an ensemble of policies distilled on a diverse dataset can generalise significantly better than the original agent.

## 1 Introduction

A major challenge for developing reliable reinforcement learning (RL) agents is their ability to generalise to new scenarios they did not encounter during training. The zero-shot policy transfer setting (ZSPT, Kirk et al., 2023) tests for this ability by having an agent train on a fixed set of training environments, referred to as training *contexts*, and measuring the agent's performance on a held-out set of similar, but different, testing contexts. Previous work has identified that *policy distillation* after training, the act of transferring knowledge from the agent's policy into a freshly initialised neural network, can be used as a tool for generalisation. In particular, it has been shown that the distilled policy sometimes achieves higher test performance than the original policy (Lyle et al., 2022).

However, it is not yet entirely clear *how* policy distillation after training can improve generalisation performance in RL. Lyle et al. (2022) theoretically show that the temporal difference (TD) loss negatively affects the smoothness of the learned value function, which only indirectly explains why policy distillation after training (without TD loss) can improve generalisation. They also partially attribute the observed generalisation benefits to the stationarity of the distillation targets, which avoids negative effects induced by the non-stationary RL targets during training (Igl et al., 2021), but this lacks a solid theoretical justification. Moreover, recent work has shown that not only the stationarity of the RL training distribution, but also its overall diversity can affect generalisation to unseen contexts (Jiang et al., 2023; Suau et al., 2024; Weltevrede et al., 2025). This additionally

raises the question whether we can increase generalisation performance by changing the distribution of data on which the policy is distilled.

In this paper, we theoretically analyse the act of distilling a policy after training, and try to answer *how* the policy should be distilled and on *what* data. Our analysis is based on the idea that many real-world data distributions exhibit symmetries, and that generalising to novel inputs will require being invariant to those symmetries. Although there is a lot of empirical evidence that demonstrates neural networks can learn invariances from data in a wide variety of settings and applications (Shorten and Khoshgoftaar, 2019; Feng et al., 2021; Zhang et al., 2021), proving this often requires stricter assumptions. Therefore, we analyse policy distillation in a *generalisation through invariance* ZSPT (GTI-ZSPT) setting, in which the agent has to learn invariance to a symmetry group, whilst only observing a subgroup of those symmetries during training. For this setting, we prove a generalisation bound for a distilled policy, and deduce insights that should translate beyond the strict group theoretical framework required for the theory.

Specifically, our theoretical results lead to two practical insights: generalisation performance can be improved by 1) training an ensemble of distilled policies, and 2) distilling on a diverse set of states. Training an ensemble can be very costly. However, we demonstrate that generalisation can be improved (at only a fraction of the sample cost required for training the RL agent), by instead creating an ensemble *after* training by distilling the agent several times and averaging the resulting policy. Finally, related to our work on policy distillation, recent work has suggested that the generalisation performance of behaviour cloning (BC) is competitive with state-of-the-art offline RL in the ZSPT setting (Mediratta et al., 2024). We demonstrate the insights for policy distillation also transfer to the BC setting and produce better generalising behaviour cloned policies. Our contributions are:

- Given a policy (for example, an RL agent after training), we prove a bound on the test performance for a distilled policy in the GTI-ZSPT setting. This bound is improved by 1) distilling a larger ensemble of policies, and 2) distillation over a more diverse set of states.
- Inspired by the theoretical results, we empirically show that the insights gained from the theory improve generalisation of behaviour cloned and distilled policies in more general settings, when the strict assumptions required for the theory no longer hold. Furthermore, we demonstrate that an ensemble of policies distilled on a diverse dataset can generalise significantly better than the original RL agent.

## 2   Background

The goal in reinforcement learning is to optimise a decision-making process, usually formalised as a Markov decision-making process (MDP) defined by the 6 tuple $\mathcal{M} = (S, A, T, R, p_0, \gamma)$. In this tuple, $S$ denotes the state space, $A$ the action space, $T : S \times A \to \mathcal{P}(S)$ the transition model, $R : S \times A \to \mathbb{R}$ the reward function, $p_0 : \mathcal{P}(S)$ the initial state distribution and $\gamma \in [0, 1]$ the discount factor, where $\mathcal{P}(S)$ denotes the probability function over state space $S$. Optimising an MDP corresponds to finding the policy $\pi : S \to \mathcal{P}(A)$ that maximises the return (the expected discounted sum of rewards) $J^\pi = \mathbb{E}_\pi[\sum_{t=0}^\infty \gamma^t r_t]$. The expectation here is over the Markov chain $\{s_0, a_0, r_0, s_1, a_1, r_1, ...\}$ induced by following policy $\pi$ in MDP $\mathcal{M}$ (Akshay et al., 2013). The optimal policy $\pi^* = \arg\max_\pi \mathbb{E}_\pi[\sum_{t=0}^\infty \gamma^t r_t]$ is the policy that maximises this return. The *on-policy* distribution $\rho_\mathcal{M}^\pi : \mathcal{P}(S)$ is the distribution over the states that a policy $\pi$ would visit in an MDP $\mathcal{M}$.

A contextual Markov decision-making process (CMDP) $\mathcal{M}|_C$ (Hallak et al., 2015) is an MDP where the state space $S = S' \times C$ can be structured as an outer product of a context space $C$ and underlying state space $S'$. A context $c \in C$ is sampled at the start of an episode and does not change thereafter. The context is part of the state and can influence the transitions and rewards. As such, it can be thought of as defining a task or specific environment that the agent has to solve in that episode. In the *zero-shot policy transfer* (ZSPT) setting (Kirk et al., 2023), an agent gets to train on a fixed subset of contexts $C_{train} \subset C$ and has to generalise to a distinct set of testing contexts $C_{test} \subset C, C_{train} \cap C_{test} = \varnothing$. In other words, the agent gets to interact and train in the CMDP $\mathcal{M}|_{C_{train}}$ (the CMDP induced by the training contexts $C_{train}$), but has to maximise return in the testing CMDP $\mathcal{M}|_{C_{test}}$.

### 2.1   Policy distillation

In policy distillation, a knowledge transfer occurs by distilling a policy from a *teacher* network into a newly initialised *student* network. There are many different ways the policy can be distilled

([Czarnecki et al., 2019](#)), but in this paper we consider a student network that is distilled on a fixed dataset, that is collected after training, and usually (but not necessarily) consists of on-policy data collected by the teacher. For analysis, we simplify the setting by assuming a deterministic, scalar student and teacher policy $\pi_\theta : S \to \mathbb{R}, \pi_\beta : S \to \mathbb{R}$. The distillation loss we consider is simply the mean squared error (MSE) between the output of the two policies:

$$l_D(\theta, \mathcal{D}, \pi_\beta) = \frac{1}{n} \sum_{s \in \mathcal{D}} (\pi_\theta(s) - \pi_\beta(s))^2 \tag{1}$$

where $\mathcal{D} = \{s_1, ..., s_n\}$ is the set of states we distil on. This simplified distillation setting is only used for the theoretical results, our experiments in Section 5 consider more general settings. Note, we consider *behaviour cloning* (BC) as a specific instance of policy distillation, where the student network only has access to a fixed dataset of the teacher's behaviour (state-action tuples). For more on behaviour cloning, distillation and their differences, we refer to Appendix A.1.

If we assume a certain smoothness of the transitions, rewards and policies (in particular, Lipschitz continuous MDP and policies), it is possible to bound the performance difference between the student and an optimal policy ([Maran et al., 2023](#), Theorem 3):

**Theorem 3.** *Let $\pi^*$ be the optimal policy and $\pi_\theta$ be the student policy. If the MDP is $(L_T, L_R)$-Lipschitz continuous and the optimal and student policies are $L_\pi$-Lipschitz continuous, and we have that $\gamma L_T(1 + L_\pi) < 1$, then it holds that:*

$$J^{\pi^*} - J^{\pi_\theta} \leq \frac{L_R}{(1 - \gamma)(1 - \gamma L_T(1 + L_{\pi_\theta}))} \mathbb{E}_{s \sim d^{\pi^*}} [\mathcal{W}(\pi^*(\cdot|s), \pi_\theta(\cdot|s))]$$

*where $d^{\pi^*}(s) = (1 - \gamma) \sum_{t=0}^\infty \gamma^t \mathbb{P}(s_t = s|\pi^*, p_0)$ the $\gamma$-discounted visitation distribution.*

*Proof.* See Appendix E.1 for the proof and exact definitions of all the terms. $\square$

In other words, under these conditions the return of a student policy can be bounded by the distance between the student and optimal policies along the states visited by the optimal policy.

## 2.2 Symmetry groups

To formalise the notion of symmetries and invariance of a function $f : X \to Y$, it is useful to define a *symmetry group* $G$. A group is a non-empty set $G$ together with a binary operation $\cdot$ that satisfies certain requirements such as closure, associativity and always containing an inverse and identity element.[1] A group and its elements are abstract mathematical notions. In order to apply elements from a group to a vector space $X$, we need to define a *group representation* $\psi_X$ that maps group elements to invertible matrices. In this paper, we always assume the representations are orthogonal (i.e. $\psi_X(g^{-1}) = \psi_X(g)^\top$). Note that we only need to define the representations $\psi_X$ for the analysis, as they are part of the generalisation bound but are not explicitly defined for the experiments.

We can define the invariance of a function $f$ as

$$f(\psi_X(g)x) = f(x) \quad \forall x \in X, g \in G. \tag{2}$$

A subset $B$ of $G$ is called a *subgroup* of $G$ (notation $B \leq G$) if the members of $B$ form a group themselves. Any group $G$ has at least one subgroup, the *trivial subgroup*, consisting of only the identity element $e : e \circ g = g \circ e = g, \forall g \in G$. A subgroup $B \leq G$ is finite if it has a finite number of elements. For more on group theory, we refer to Appendix A.2.

**Example** The group $SO(2)$ consists of the set of all rotations in two dimensions. If the input to function $f$ consists of Euclidean coordinates $(x, y)$, the group representation $\psi_X$ maps a rotation of $\alpha$ degrees to the 2D rotation matrix associated with an $\alpha$ degree rotation. The function $f$ would be considered rotationally invariant if $f(\psi_X(\alpha)x) = f(x), \forall x \in \mathbb{R}^2, \alpha \in SO(2)$. An example of a finite subgroup of $SO(2)$ is the group $C_4$ consisting of all $90°$ rotations, or the subgroup consisting of only the identity element ($0°$ rotation).

---

[1] In this paper, we abuse notation slightly by denoting both the group and the non-empty set with $G$, depending on context.

One approach to induce a function that is invariant to the symmetry group $G$ is to train it with *data augmentation*. For groups of finite size, it is possible to perform *full data augmentation*, which consists of applying every transformation in $G$, to each element of an original dataset $\mathcal{T} = (\mathcal{X}, \mathcal{Y}) = \{(x, y) \in X, Y\}^n$. The function is then trained on the *augmented* dataset $\mathcal{T}_G = \{(\psi_X(g)x, y) | \forall (x, y) \in \mathcal{T}, g \in G\}$. In general, training a function with data augmentation does not guarantee it becomes invariant (Flinth and Ohlsson, 2023), it instead can become approximately invariant or invariant only on the distribution of data on which it was trained (Kvinge et al., 2022; Lyle et al., 2020; Azulay and Weiss, 2019). However, under certain conditions, the average of an infinitely large ensemble *can* have that guarantee (Gerken and Kessel, 2024; Nordenfors and Flinth, 2024).

## 2.3 Ensembles and invariance

Formally, an ensemble consists of multiple neural networks $f_\theta : X \to \mathbb{R}$ with parameters $\theta \sim \mu$ initialised from some distribution $\mu$ and trained on the same dataset $\mathcal{T} = (\mathcal{X}, \mathcal{Y}) = \{(x, y) \in X, Y\}^n$. The output of an infinitely large ensemble $\bar{f}_t(x)$ at training time $t$ is given by the average over the *ensemble members* $f_\theta$: $\bar{f}_t(x) = \mathbb{E}_{\theta \sim \mu}[f_{\mathcal{L}_t\theta}(x)]$, where $\mathcal{L}_t$ denotes a map from initial parameters $\theta$ to the corresponding parameters after $t$ steps of gradient descent. In practice, the infinite ensemble is approximated with a finite Monte Carlo estimate of the expectation $\hat{f}_t$: $\bar{f}_t \approx \hat{f}_t = \frac{1}{N} \sum_{i=1}^{N} f_{\mathcal{L}_t\theta_i}(x)$, where $\theta_i \sim \mu$ and $N$ is the size of the ensemble.

### 2.3.1 Infinite width limit

Although there does not yet exist a single comprehensive theoretical framework for how neural networks work, significant progress has been made in the field of deep learning theory in the limit of infinite layer width. In this limit, an infinite ensemble $\bar{f}_t(x)$ trained with MSE loss follows a simple Gaussian distribution that depends on the network architecture and initialisation (Jacot et al., 2018; Lee et al., 2019). Gerken and Kessel (2024) prove that the infinite ensemble $\bar{f}_t(x)$ trained on the augmented dataset $\mathcal{T}_G$ for some group $G$, satisfies the definition of invariance in equation (2), for any $t$ and any $x$. In other words, an infinitely large ensemble of infinitely wide neural networks, trained with full data augmentation for group $G$, is invariant under transformations from $G$ for any input and at any point during the training process. In our analysis, we use Lemma 6.2 from Gerken and Kessel (2024) that bounds the invariance of an infinite ensemble of infinitely wide networks trained with full data augmentation on a finite subgroup $B \leq G$:

**Lemma 6.2.** *Let $\bar{f}_t(x) = \mathbb{E}_{\theta \sim \mu}[f_{\mathcal{L}_t\theta}(x)]$ be an infinite ensemble of neural networks with Lipschitz continuous derivatives with respect to the parameters. Define the error $\kappa$ as a measure of discrepancy between representations from the group $G$ and its finite subgroup $B$:*

$$\kappa = \max_{g \in G} \min_{b \in B} ||\psi_X(g) - \psi_X(b)||_{op} \tag{3}$$

*where $|| \cdot ||_{op}$ denotes the operator norm. The prediction of an infinite ensemble trained with full data augmentation on $B \leq G$ deviates from invariance by*

$$\left| \bar{f}_t(x) - \bar{f}_t(\psi_X(g)x) \right| \leq \kappa\, C(x), \qquad \forall g \in G \tag{4}$$

*for any time $t$. Here $C$ is a function of $x$ independent of $g$.*

*Proof.* See Appendix E.2 for the proof and exact definitions of all the terms. $\square$

This lemma bounds the deviation from invariance of the infinite ensemble by a factor $\kappa$, which is a measure of how well the subgroup $B$ covers the full group $G$, in the space of representations $\psi_X$. For more background on infinite ensembles in the infinitely wide limit, we refer to Appendix A.3.

## 3 Related work

The CMDP framework captures many RL settings focused on zero-shot generalisation (Kirk et al., 2023). Some approaches to improve generalisation focus on learning generalisable functions through inductive biases (Kansky et al., 2017; Wang et al., 2021) or by applying regularisation techniques from supervised learning (Tishby and Zaslavsky, 2015; Cobbe et al., 2019). These approaches improve

generalisation by changing the RL training process, whereas we distil a teacher policy *after* training, which in principle is agnostic to how that teacher was trained. Other work improves generalisation by increasing the diversity of the data on which the agent trains, for example by increasing the diversity of the training contexts using domain randomisation (Tobin et al., 2017; Sadeghi and Levine, 2017), or creating artificial data using data augmentation (Lee et al., 2020; Raileanu et al., 2021). Our work focusses on sampling additional data from a fixed set of training contexts, but differs from data augmentation in that we do not require explicitly designed augmentations. For a broader survey on zero-shot generalisation in reinforcement learning, see Kirk et al. (2023).

Policy distillation in RL has been used to compress policies, speed up learning, or train multi-task agents by transferring knowledge from teacher policies to student networks (Rusu et al., 2016; Schmitt et al., 2018; Czarnecki et al., 2019). Various methods of distillation exist, balancing factors such as teacher quality, access to online data, and availability of teacher value functions or rewards (Czarnecki et al., 2019). Some studies have used distillation to improve generalisation, either by mitigating RL-specific non-stationarity through periodic distillation (Igl et al., 2021) or by distilling from policies trained with privileged information or weak augmentations (Fan et al., 2021; Walsman et al., 2023). Most similar to our work, Lyle et al. (2022) show a policy distilled after training can sometimes generalise better than the original RL agent. But, their theory only indirectly covers policy distillation and they do not investigate how the distillation data affects generalisation.

## 4 Generalisation through invariance

In this section, we introduce a specific ZSPT setting that allows us to prove a generalisation bound for a distilled policy. The main idea is that many real-world data distributions exhibit symmetries, and that generalising to novel inputs sampled from this distribution requires (at least partially) being invariant to those symmetries. Moreover, any training dataset sampled IID from this distribution will likely observe some of these symmetries.

Proving a neural network learns invariances from data is not straightforward, and usually requires assumptions on the mathematical structure of the symmetries. For this reason, we consider a specific setting in which an agent has to become invariant to a symmetry group $G$, but trains with full data augmentation under only a subgroup $B \leq G$. Even though this setting requires strict assumptions, we expect the insights to apply more broadly, as there is a lot of empirical evidence that data augmentation improves generalisation performance in a wide variety of settings and applications (Shorten and Khoshgoftaar, 2019; Feng et al., 2021; Zhang et al., 2021; Miao et al., 2023). We formalise the idea in a *generalisation through invariance ZSPT* (GTI-ZSPT)

**Definition 1** (Generalisation through invariance ZSPT). *Let $\mathcal{M}|_C$ be a CMDP and let $C_{train}, C_{test} \subset C$ be a set of training and testing contexts that define a ZSPT problem. Additionally, let $\pi^*$ be the optimal policy in $\mathcal{M}|_C$, $S_{\mathcal{M}|_C}^{\pi^*} = \{s \in S | \rho_{\mathcal{M}|_C}^{\pi^*}(s) > 0\}$ denote the set of states with non-zero support under the on-policy distribution $\rho_{\mathcal{M}|_C}^{\pi^*}$ in CMDP $\mathcal{M}|_C$. In the generalisation through invariance ZSPT (GTI-ZSPT), the sets $S_{\mathcal{M}|_C}^{\pi^*}$ and $S_{\mathcal{M}|_{C_{train}}}^{\pi^*}$ admit a symmetric structure:*

$$S_{\mathcal{M}|_C}^{\pi^*} = \{\psi_S(g)s | g \in G, s \in \bar{S}\}$$
$$S_{\mathcal{M}|_{C_{train}}}^{\pi^*} = \{\psi_S(b)s | b \in B, s \in \bar{S}\}, \quad B \leq G$$

*where $\bar{S} \subset S_{\mathcal{M}|_{C_{train}}}^{\pi^*}$ is a proper subset of $S_{\mathcal{M}|_{C_{train}}}^{\pi^*}$ and $G$ is a non-trivial symmetry group (and $B \leq G$ a finite subgroup) that leaves the optimal policy invariant: $\pi^*(s) = \pi^*(\psi_S(g)s), \forall s \in \bar{S}$.*

To quantify the discrepancy between the group and its subgroup, the following measure is defined (Gerken and Kessel, 2024):

**Definition 2.** *For the group $G$ and its finite subgroup $B \leq G$ that define the symmetric structure of a GTI-ZSPT (Definition 1), $\kappa$ is a measure of discrepancy between the representations of these groups:*

$$\kappa = \max_{g \in G} \min_{b \in B} ||\psi_S(g) - \psi_S(b)||_{op}$$

*where $|| \cdot ||_{op}$ denotes the operator norm.*

The constant $\kappa$ measures how much the subgroup $B \leq G$ deviates from the full group $G$. The bigger the subgroup $B$ is, the smaller $\kappa$ will become (with $\kappa = 0$ in the limit of $B = G$).

**Example** The 'Reacher with rotational symmetry' ZSPT in Figure 1 satisfies the conditions of the GTI-ZSPT. This is a continuous control environment where the agent has to move a robot arm (blue) in such a way that its hand (black circle) reaches the goal location (green circle). The four training contexts have shoulder locations that are rotated 0, 90, 180 and 270 degrees around the goal location. In testing, the shoulder can be rotated any amount. As an example, the measure $\kappa$ for the subgroup $C_4$ of $90°$ rotations (as depicted in the figure), would be larger than for the bigger subgroup $C_8$ of $45°$ rotations. See Appendix C.1 for more on this example and how it satisfies the assumptions.

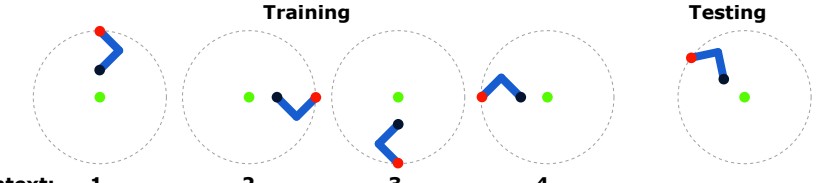

**Training**          **Testing**

**Context:**   **1**        **2**        **3**        **4**

Figure 1: A 'Reacher with rotational symmetry' CMDP with four training contexts, differing in the location of the shoulder (red), positioned along a circle (dotted line). All contexts share the relative pose of the robot arm (blue). The goal is for the hand (black circle) to reach the goal location (green circle) in the middle. The training contexts can be generated by applying the group of $90°$ rotations to context 1, and the testing contexts can be generated with the full group of rotations ($SO(2)$).

## 4.1 Bounding the performance

For the GTI-ZSPT setting, we can bound the performance of a distilled policy in the testing CMDP with the following theorem:

**Theorem 1.** *Consider policy distillation for a deterministic, scalar teacher policy $\pi_\beta : S \to \mathbb{R}$ (Equation (1) in Section 2.1) in a $L_T, L_R$-Lipschitz continuous CMDP in the GTI-ZSPT setting. Let the student policy $\hat{\pi}_\infty$ be an ensemble of $N$ infinitely wide neural networks $\pi_\theta : S \to \mathbb{R}$ with Lipschitz continuous derivatives with respect to its parameters, distilled on an on-policy dataset $\mathcal{D} = S_{\mathcal{M}|_{C_{train}}}^{\pi_\beta} = \{\psi_S(b)s | b \in B, s \in \bar{S}\}$ consisting of all the states in the training contexts encountered by the teacher in the GTI-ZSPT setting. Furthermore, let the student policy be $L_{\hat{\pi}_\infty}$- Lipschitz continuous and assume $\gamma L_T(1 + L_{\hat{\pi}_\infty}) < 1$.*

*If the teacher is optimal in the training tasks $C_{train}$ (but arbitrarily bad anywhere else), the performance of the student in the testing CMDP $\mathcal{M}|_{C_{test}}$ is bounded with probability at least $1 - \epsilon$, by:*

$$J^{\pi^*} - J^{\hat{\pi}_\infty} \leq \frac{L_R}{(1-\gamma)(1-\gamma L_T(1+L_{\hat{\pi}_\infty}))}\left(\kappa \bar{C}_\Theta + \frac{1}{\sqrt{N}}\bar{C}_{\Sigma_\infty}(\epsilon)\right) \tag{5}$$

*where $\kappa$ is the measure of discrepancy between subgroup $B \leq G$ and full group $G$ (see definition 2) and $\bar{C}_\Theta, \bar{C}_{\Sigma_\infty}$ are constants that depend on the $\gamma$-discounted visitation distribution of the optimal policy in $\mathcal{M}|_{C_{test}}$, the network architecture, and the dataset $\mathcal{D}$. Additionally, $\bar{C}_{\Sigma_\infty}$ also depends on the network initialisation and the confidence level $\epsilon$.*

*Proof.* Thanks to the symmetric structure of the GTI-ZSPT, we can bound the output of an infinite ensemble of distilled policies $\bar{\pi}_\infty$, when evaluated on testing states in $\mathcal{M}_{C_{test}}$, using the bound on the deviation from invariance from Section 2.3.1. This can be combined with a probabilistic bound for Monte Carlo estimators to bound the output of a finite ensemble $\hat{\pi}_\infty$ on the testing states. With this bound on the output of the student policy, we can use the performance bound for Lipschitz continuous MDPs from Section 2.1 to get our final result above. See Appendix B for the full proof.   □

The theorem above offers two insights:

1. The bigger the ensemble size $N$, the smaller the bound on performance.
2. The bigger the subgroup $B$, the smaller the measure $\kappa$, the smaller the bound on performance.

As we mentioned before, even though Theorem 1 requires strict assumptions, we believe the insights apply more broadly. Essentially, the theorem relies on the generalisation benefits induced by training on additional samples generated by performing data augmentation. In practice, it often doesn't matter if the augmentations form a group, are consistent with the original data distribution, or are applied to all classes equally (Bishop, 1995; Wu et al., 2020; Hansen and Wang, 2021; Lin et al., 2022; Geiping et al., 2023; Miao et al., 2023). As such, we believe that in many settings, the benefits of training on a bigger subgroup $B$, can also be realised by simply training on more diverse data, which we clarify with some examples in our experiments.

## 5 Experiments

In this section, we demonstrate that the insights provided by the theory translate to practical and workable principles that can improve the generalisation performance of a distilled policy, beyond the performance of the original agent. In Section 5.1, we establish that bigger ensembles indeed improve generalisation and show what it means to train on a bigger subgroup $B \leq G$ in the illustrative CMDP from Figure 1. This experiment satisfies the assumptions for the GTI-ZSPT setting, but does not strictly satisfy some of the non-practical assumptions required for Theorem 1. In Section 5.2, we demonstrate that the insights also apply to the more complex Minigrid Four Rooms environment (Chevalier-Boisvert et al., 2023) that breaks most of the assumptions required for the proof in Section 4. For experimental details, see Appendix C.

### 5.1 Reacher with rotational symmetry

Table 1: Performance of distilled policies in the Illustrative CMDP from Figure 1 for different ensemble sizes $N$ (trained under subgroup $B = C_4$) and different subgroups $B \leq SO(2)$ (for $N = 1$). Shown are the mean and standard deviation for 20 seeds, and in bold are the best returns including those with overlapping 95% confidence intervals.

| Ensemble Size $N$: | N=1 | N=10 | N=100 |
|---|---|---|---|
| Train Performance | **1.17 ± 0.004** | **1.17 ± 0.004** | **1.17 ± 0.003** |
| Test Performance | 0.75 ± 0.147 | 0.89 ± 0.107 | **1.05 ± 0.117** |
| **Subgroup $B \leq SO(2)$:** | $B = C_2$ | $B = C_4$ | $B = C_8$ |
| Train Performance | **1.17 ± 0.003** | **1.17 ± 0.004** | 1.16 ± 0.002 |
| Test Performance | 0.39 ± 0.0805 | 0.75 ± 0.147 | **1.11 ± 0.072** |

The theory proves that we can reduce an *upper bound* on the difference to optimal performance when we increase the ensemble size $N$ and train on a bigger subgroup $B \leq G$. However, that does not always guarantee strict performance improvements (for example, if the upper bound were so large it is meaningless). Furthermore, the theory requires some assumptions that are not always practical, such as infinitely wide networks, scalar-valued policies, or a Lipschitz-continuous reward function, that we do not expect to affect the overall result in practice. Therefore, we investigate whether the insights from Section 4 hold without these assumptions, and whether they lead to actual generalisation improvements. In Table 1 we show that increasing the size of the ensemble, consisting of networks of finite width, *does* actually lead to higher test performance in the CMDP from Figure 1.

Additionally, in Table 1 we show that generalisation performance is affected by the size of the subgroup $B \leq SO(2)$ we train on. Only training on two training contexts, corresponding to the subgroup $C_2 \leq SO(2)$ of 180° rotations, performs worse than training on four contexts, corresponding to the subgroup $C_4 \leq SO(2)$ of 90° rotations (as shown in Figure 1). Furthermore, training on eight contexts (subgroup $C_8 \leq SO(2)$ of all 45° rotations) is even better. In this illustrative CMDP, training on larger subgroups requires training in new contexts, but this is not always the case.

#### 5.1.1 Improving generalisation with diverse data from the same contexts

In sufficiently complex CMDPs, there are several dimensions of variation between different contexts. For example, we can add different starting poses to the contexts in the CMDP from Figure 1, such that a context is now defined by a rotation of the shoulder *and* the relative pose of the robot arm

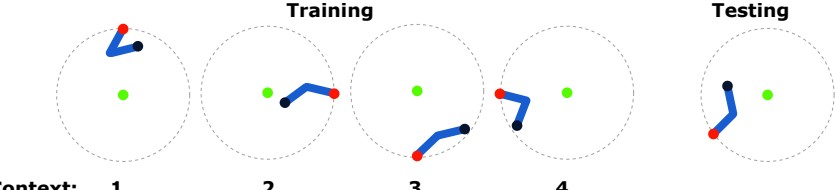

Figure 2: The base context set in the illustrative reacher CMDP with varying shoulder location (red) *and* robot arm pose (blue), see Figure 1 for details.

(see Figure 2). This CMDP does not strictly satisfy the symmetry conditions in Definition 1, but invariance to rotations is still a major component for generalisation. Since the training contexts now also differ in the starting pose, the dataset generated from the training contexts no longer corresponds to performing full data augmentation with respect to a rotational symmetry. However, in this example, this can be fixed by training on additional data from the given set of training contexts.

To illustrate this, we compare three distillation datasets:

1. **Training Contexts:** This dataset consists of the teacher's trajectories in the training contexts (contexts 1 through 4 in Figure 2).
2. **Training Contexts + $C_4$:** This dataset consists of the Training Contexts dataset, but with additional trajectories sampled from different starting poses in the same contexts. In particular, for each context, it includes trajectories starting in the rotated poses from the other contexts. This dataset corresponds to performing full data augmentation for the 90° rotations subgroup $C_4$ on the Training Contexts dataset (see Appendix C.1 for a visual representation of this).
3. **Training Contexts + Random:** Like Training Contexts + $C_4$, this dataset includes additional trajectories from different starting poses. However, for this dataset the new starting poses are sampled uniformly at random.

The Training Contexts + $C_4$ dataset illustrates how in this CMDP the subgroup $B \leq SO(2)$ can be increased by sampling additional trajectories from the same training contexts (technically, the Training Contexts dataset corresponds to the trivial subgroup $\{e\} \leq SO(2)$ consisting of only the identity element $e$, which is smaller than $C_4 \leq SO(2)$). In Table 2, we see that training on this dataset indeed produces higher test performance than the Training Contexts dataset. However, the same generalisation benefits are also observed for the Training Contexts + Random dataset.

Table 2: Performance of distilled policies (for $N = 1$) in the Illustrative CMDP from Figure 2 for different datasets. The datasets consist of the teacher's trajectories sampled for several starting states. Shown are the mean and standard deviation for 20 seeds, and in bold are the best returns including those with overlapping 95% confidence intervals.

| **Distillation Dataset** | Train | Test |
|---|---|---|
| Training Contexts | **1.20 $\pm$ 0.157** | 0.39 $\pm$ 0.051 |
| Training Contexts + $C_4$ | **1.11 $\pm$ 0.099** | **0.48 $\pm$ 0.080** |
| Training Contexts + Random | **1.14 $\pm$ 0.072** | **0.49 $\pm$ 0.077** |

The Training Contexts + Random dataset illustrates that the generalisation benefits of data augmentation go far beyond the "training to be invariant under a group symmetry" paradigm. Some studies suggest that the benefits are simply due to the regularising effect that data augmentation can provide (Bishop, 1995; Wu et al., 2020; Hansen and Wang, 2021), or by making it more difficult to overfit to spurious correlations (Raileanu et al., 2021; Shen et al., 2022). In this sense, we expect the insight of training with full data augmentation on a bigger subgroup $B \leq G$ from Theorem 1, to translate in practice to simply training on more diverse data, even data that is sampled from the same contexts.

## 5.2 Four Rooms

In this section, we demonstrate that increasing ensemble size and data diversity can significantly increase the generalisation performance of a distilled policy, even when most of the assumptions for Theorem 1 no longer hold. The Four Rooms grid world environment from the Minigrid benchmark

does not appear to have an invariant symmetry that plays a core part in generalising to new contexts, as required for the definition of a GTI-ZSPT. The teacher is an agent trained with Proximal Policy Optimisation (PPO Schulman et al., 2017) and is therefore not necessarily optimal in the training contexts. Additionally, the teacher is a stochastic policy that is distilled by regressing on the vector of probabilities or behaviour cloned using a logarithmic loss (see Appendix A.1 for more background on these losses).

### 5.2.1 Policy distillation improves generalisation

For the experiments in the Four Rooms environment the teacher is a policy trained with the PPO+Explore-Go algorithm for 8 million environment steps. The Explore-Go approach was introduced by Weltevrede et al. (2025) to increase generalisation by generating a more diverse training distribution for the RL agent. It leverages a separately trained pure exploration agent, rolled out at the beginning of each episode, to artificially increase the starting state distribution for the PPO agent.[2] Since this teacher trains on a more diverse state distribution than a normal PPO agent, it provides good teaching targets for our distillation datasets. We compare the following three datasets:

1. **Teacher:** This dataset consists of the teacher's trajectories in the (original) training contexts.
2. **Explore-Go:** This dataset mimics the training distribution for the Explore-Go approach by sampling teacher trajectories from additional starting states, generated by a pure exploration policy rolled out at the start of each episode. This dataset has the property that all the data is on-policy for our teacher, yet more diverse than the Teacher dataset.
3. **Mixed:** This dataset is a 50/50 mix of Teacher and trajectories collected by a separately trained pure exploration policy. This dataset is diverse, but does not solely consist of states encountered by the teacher.

In Table 3 we can see that the more diverse datasets (Mixed and Explore-Go) significantly outperform the Teacher dataset and that the ensemble of size $N = 10$ outperforms the single student $N = 1$ for each dataset type. Moreover, the ensemble, distilled on the Explore-Go dataset, generalises significantly better than the original PPO agent, whilst only requiring around 12% additional environment steps (compared to the teacher's training budget).

Table 3: Performance of an ensemble (of size $N$) of policy distillation or behaviour cloning policies on various datasets compared to the PPO+Explore-GO teacher in the Four Rooms environment. Shown are mean and standard deviation over 20 seeds, and in bold are the best returns including those with overlapping 95% confidence intervals (within the same category).

|  | Dataset | Train (N=1) | Train (N=10) | Test (N=1) | Test (N=10) |
|---|---|---|---|---|---|
| **PPO+Explore-Go** | - | $\mathbf{0.92 \pm 0.020}$ | - | $\mathbf{0.74 \pm 0.040}$ | - |
| **Distillation** | Teacher | $\mathbf{0.92 \pm 0.020}$ | $\mathbf{0.92 \pm 0.020}$ | $0.56 \pm 0.049$ | $0.67 \pm 0.054$ |
|  | Mixed | $\mathbf{0.92 \pm 0.020}$ | $\mathbf{0.92 \pm 0.020}$ | $0.72 \pm 0.040$ | $0.84 \pm 0.034$ |
|  | Explore-Go | $\mathbf{0.92 \pm 0.020}$ | $\mathbf{0.92 \pm 0.019}$ | $0.78 \pm 0.041$ | $\mathbf{0.88 \pm 0.036}$ |
| **Behaviour Cloning** | Teacher | $\mathbf{0.91 \pm 0.022}$ | $\mathbf{0.92 \pm 0.020}$ | $0.26 \pm 0.046$ | $0.37 \pm 0.054$ |
|  | Mixed | $0.86 \pm 0.031$ | $\mathbf{0.91 \pm 0.025}$ | $0.15 \pm 0.024$ | $0.20 \pm 0.026$ |
|  | Explore-Go | $0.87 \pm 0.028$ | $\mathbf{0.92 \pm 0.021}$ | $0.56 \pm 0.060$ | $\mathbf{0.75 \pm 0.045}$ |

Lastly, we demonstrate in this section that the same insights also hold for a logarithmic behaviour cloning loss for stochastic policies that is widely used in practice (Foster et al., 2024). At the bottom of Table 3, we show that the an ensemble (of size $N = 10$), distilled on the Explore-Go dataset, generalises significantly better than a single behaviour cloning agent on the Teacher dataset. Note that behaviour cloning achieves lower performances than distillation, and that BC performs considerably worse on the Mixed dataset. In our definition of behaviour cloning, the student policy learns to imitate whatever policy collected the dataset, by only observing the actions that were actually sampled during collection. Therefore, the BC agent performs worse than the distillation agent, since the latter has access to more information (all the action probabilities of the teacher). On the Mixed dataset, the BC agent clones the behaviour policy that consists of a 50/50 mix of the (optimal) Teacher policy and (suboptimal) pure exploration policy. The resulting cloned behaviour performs even worse than

---

[2]For pure exploration, the objective focuses solely on exploring new parts of the state space, ignoring rewards.

the BC agent trained on the Teacher dataset. In contrast, the policy distillation agent on the Mixed dataset regresses on the action probabilities of the Teacher, on the states encountered by the 50/50 mixture of policies, and therefore has a much better learning target.

# 6 Discussion and limitations

The experiments in the Four Rooms environment in Section 5.2.1 serve to empirically demonstrate how our insights can be leveraged to significantly enhance the generalisation performance of a reinforcement learning agent through policy distillation. A clear example of this is seen in our ensemble $N = 10$, distilled on the Explore-Go dataset, which achieves substantially higher test performance than the original PPO+Explore-Go teacher policy (see Table 3). The potential of policy distillation after training as a tool to improve generalisation was initially identified in Lyle et al. (2022), but we believe the results of this paper provide a more compelling argument and empirical evidence for this phenomenon.

Whether the benefits of performing data augmentation with respect to some symmetry group actually stem from induced invariance or reduced overfitting and other forms of regularisation, is still an ongoing topic of discussion in the literature (Lyle et al., 2020; Shen et al., 2022). To add to this discussion, in Appendix D.1 we measure the invariance of our trained models on the 'Reacher with rotational symmetry' experiments from Table 1 and plot it against the ensemble size $N$ and subgroup $B \leq SO(2)$. We find that in this particular experiment, the distilled policies *do* become more invariant as ensemble and subgroup size increase, just as our theory predicts.

Obtaining tight generalisation bounds for neural networks is notoriously challenging (Jiang et al., 2020; Gastpar et al., 2024). Moreover, some of the assumptions for Theorem 1, such as infinitely wide neural networks, are hard to meet in practice. Therefore, we believe the true strength of our theory lies in its ability to identify crucial properties of the dataset distribution and distilled ensemble that are capable of improving generalisation performance. Nonetheless, in Appendix D.2, we analyse how well our results fit the $a + \frac{b}{\sqrt{N}}$ relation identified by our theory. We find that our results reasonably agree with the shape of the theoretical upper bound, suggesting that our bound is not completely vacuous.

Finally, all ensemble members are trained independently, and during inference, can also be evaluated independently (an independent forward pass with an average over the output of the ensemble afterwards). This inherent independence means that both the training and inference processes of the ensemble are parallelisable. If parallelisation is not feasible, the runtime for both training and inference would increase linearly with the ensemble size. It is important to note that all ensemble members are distilled on the same dataset. This means the small number of additional environment steps required to sample this dataset is independent of ensemble size.

# 7 Conclusion

In this paper, we investigate the advantage of policy distillation for improving zero-shot policy transfer (ZSPT) in reinforcement learning. We introduce the generalisation through invariance ZSPT setting, to prove a generalisation bound for a policy distilled after training. Our analysis highlights two practical insights: to 1) distil an ensemble of policies, and to 2) distil it on a diverse set of states from the training contexts. We empirically evaluate that the insights hold in the Four Rooms environment from the Minigrid benchmark, even though it does not satisfy all the assumptions required for the theory, and that they also translate to the behaviour cloning setting. Moreover, we show that distilling an ensemble of policies on diverse set of states can produce a policy that generalises significantly better than the original RL agent, thus demonstrating that policy distillation can be a powerful tool to increase generalisation performance of reinforcement learning agents.

## Acknowledgments and Disclosure of Funding

We thank Caroline Horsch, Laurens Engwegen and Oussama Azizi for fruitful discussions and feedback. The project has received funding from the EU Horizon 2020 programme under grant number 964505 (Epistemic AI) and was also partially funded by the Dutch Research Council (NWO) project *Reliable Out-of-Distribution Generalization in Deep Reinforcement Learning* with project number OCENW.M.21.234. The computational resources for empirical work were provided by the Delft AI Cluster (DAIC) (2024) and the Delft High Performance Computing Centre (DHPC) (2024).

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

# A Extended background

## A.1 Policy distillation & behaviour cloning

In policy distillation, a knowledge transfer occurs by distilling a policy from a *teacher* network into a newly initialised *student* network. Depending on the objective of the knowledge transfer, the student network can be smaller, the same size, or bigger than the teacher network. Moreover, there are many different ways the policy can be distilled (Czarnecki et al., 2019), depending on the specific loss function used (Ghosh et al., 2018; Teh et al., 2017), whether the student can collect additional data during distillation (Lin et al., 2017; Parisotto et al., 2016; Ross et al., 2011), or has access to additional information like rewards or a teacher's value function (Czarnecki et al., 2019).

We consider a student network with the same architecture and size as the teacher that is distilled on a fixed dataset (so without allowing additional interactions of the student with the environment). This fixed dataset is collected after training, and usually consists of on-policy data collected by the teacher itself. In this paper, we analyse a simplified setting where both the student and teacher policy are assumed to be deterministic and scalar: $\pi_\theta : S \to \mathbb{R}, \pi_\beta : S \to \mathbb{R}$. A simple distillation loss in this setting is the mean squared error (MSE) between the output of the two policies:

$$l_D(\theta, \mathcal{D}, \pi_\beta) = \frac{1}{n} \sum_{s \in \mathcal{D}} (\pi_\theta(s) - \pi_\beta(s))^2$$

where $\mathcal{D} = \{s_1, ..., s_n\}$ is the set of states we distil on.

More generally, distillation can be performed between deterministic, vector valued student and teacher policies $\pi_\theta : S \to \mathbb{R}^d, \pi_\beta : S \to \mathbb{R}^d$ with the loss

$$l(\theta, \mathcal{D}, \pi_\beta) = \frac{1}{n} \sum_{s \in \mathcal{D}} ||\pi_\theta(s) - \pi_\beta(s)||_2^2 \tag{6}$$

For stochastic policies, it is more common to minimise the Kullback-Leibler (KL) divergence between the student and teacher policies (Arora et al., 2018), sometimes including an entropy regularisation term (Teh et al., 2017; Lyle et al., 2022)

$$l(\theta, \mathcal{D}, \pi_\beta) = \frac{1}{n} \sum_{s \in \mathcal{D}} D_{KL}(\pi_\theta(s)||\pi_\beta(s)) + \lambda H(\pi_\theta)$$

where $H(\cdot)$ denotes the entropy of the policy. An alternative approach for discrete, stochastic policies is to regress towards the logits or probabilities over actions from the teacher:

$$l(\theta, \mathcal{D}, \pi_\beta) = \frac{1}{n} \sum_{s \in \mathcal{D}} ||\pi_\theta(\cdot|s) - \pi_\beta(\cdot|s)||_2^2 \tag{7}$$

where $\pi(\cdot|s)$ now indicates the vector (of dimension $|A|$) of probabilities or logits that policy $\pi$ produces in state $s$.

As mentioned above, usually the policy is distilled on on-policy data collected by the teacher. However, in general, a policy can in principle be distilled on any distribution over states, since the targets produced by the teacher (i.e., $\pi_\beta(s)$ or $\pi_\beta(\cdot|s)$) can be trained off-policy, independently of how the state $s$ was reached, or which action was taken in $s$ during collection.

### A.1.1 Behaviour cloning

We consider *behaviour cloning* (BC) as a specific instance of policy distillation, where the student network only has access to a fixed dataset of the teacher's behaviour (state-action tuples) and not additional information like the teacher's policy, value function or environment rewards. The goal in behaviour cloning is to learn to imitate the behaviour policy (e.g., the teacher) that collected the dataset. In this sense, it differs from the general distillation setting, in that the learning targets are always on-policy with respect to the policy (or the mixture of policies) that collected the data.

Just as for distillation, the BC loss can differ depending on whether the student policy is deterministic or stochastic. For deterministic policies, the loss is usually the MSE between the student and the action observed in the dataset:

$$l(\theta, \mathcal{D}_\beta) = \frac{1}{n} \sum_{i=0}^{n} ||\pi_\theta(s_i) - \vec{a_i}||_2^2$$

where $\mathcal{D}_\beta = \{(s_1, a_1), ..., (s_n, a_n)\}$ is a dataset of behaviour of size $n$. For stochastic policies, it is more common to use a logarithmic loss (Foster et al., 2024)

$$l_(\theta, \mathcal{D}_\beta) = -\sum_{i=0}^{n} \ln \pi_\theta(a_i | s_i) \tag{8}$$

Note that these losses mainly differ from the distillation losses in that the learning targets are the actions $a_i$ taken by the policy that collected the dataset, rather than the (potentially off-policy) teacher policy $\pi_\beta(s_i)$.

### A.2 Group Symmetry

A group is a non-empty set $G$ together with a binary operation $\cdot$ that satisfies the following requirements:

$$
\begin{aligned}
a \cdot b \in G, \quad &\forall a, b \in G & \text{(Closure)} \\
(a \cdot b) \cdot c = a \cdot (b \cdot c), \quad &\forall a, b, c \in G & \text{(Associativity)} \\
\exists e \in G, \quad e \cdot a = a \cdot e = a, \quad &\forall a \in G & \text{(Identity)} \\
\forall a \in G, \exists a^{-1} \in G, \quad a \cdot a^{-1} = a^{-1} \cdot a = e \quad & & \text{(Inverse)}
\end{aligned}
$$

We will abuse notation slightly by denoting both the group and the non-empty set with $G$, depending on context.

We can define a *group representation* $\psi_X$ acting on $X$, as a map $\psi : G \to \mathrm{GL}(X)$ from $G$ to the general linear group $\mathrm{GL}(X)$ of a vector space $X$, where the general linear group is defined as the set of $n \times n$ invertible matrices (for finite dimensional vector space $X$ with dimension $n$) with matrix multiplication as operator and where the map $\psi$ is a group homomorphism, i.e. $\psi(a)\psi(b) = \psi(a \cdot b), \quad \forall a, b \in G$. With these definitions, invariance of a function $f$ is defined as

$$f(\psi_X(g)x) = f(x) \quad \forall x \in X, g \in G$$

A useful property when performing full data augmentation with a group $G$, is that applying a transformation from $G$ to any of the training samples in $\mathcal{T}_G$, is equivalent to applying a permutation $\mathrm{p}_g$ the augmented training dataset indices:

$$\psi_X(g)x_i = x_{\mathrm{p}_g(i)}, \qquad \text{where } i \in \{1, ..., |\mathcal{T}_G|\} \tag{9}$$

### A.3 The infinite width limit

In the limit of infinite layer width, an ensemble of neural networks from random initialization follows a Gaussian process that is characterised by the neural tangent kernel (NTK Jacot et al., 2018) defined as

$$\Theta(x, x') = \sum_{l=1}^{L} \mathbb{E}_{\theta \sim \mu}\left[ \left(\frac{\partial f_\theta(x)}{\partial \theta^{(l)}}\right)^T \left(\frac{\partial f_\theta(x')}{\partial \theta^{(l)}}\right) \right],$$

where we assumed the network $f_\theta$ has $L$ layers and $\theta^{(l)}$ denotes the parameters at layers $l \in [1, L]$ respectively. The Gaussian process at time $t$ has mean $m_t$ and covariance $\Sigma_t$ (Lee et al., 2019):

$$m_t(x) = \Theta(x, x_i)[\Theta^{-1} T_t]_{ij} y_j$$
$$\Sigma_t(x, x') = \mathcal{K}(x, x') + \Sigma_t^{(1)}(x, x') - (\Sigma_t^{(2)}(x, x') + \text{ h.c.})$$

where we use the Einstein notation convention to indicate implicit sums over the dataset indices $i, j$, h.c. indicates the Hermitian conjugate of the preceding term, $T_t = (\mathbb{I} - \exp(-\eta \Theta t))$, $\mathcal{K}(x, x') = \mathbb{E}_{\theta \sim \mu}[f_\theta(x)f_\theta(x')]$ is the neural network Gaussian process (NNGP) kernel, and $\Sigma_t^{(1)}$ and $\Sigma_t^{(2)}$ are defined as follows:

$$\Sigma_t^{(1)}(x, x') = \Theta(x, x_i)[\Theta^{-1} T_t \mathcal{K} T_t \Theta^{-1}]_{ij} \Theta(x_j, x')$$
$$\Sigma_t^{(2)}(x, x') = \Theta(x, x_i)[\Theta^{-1} T_t]_{ij} \mathcal{K}(x_j, x').$$

We use shorthand notation $\Sigma_t(x, x) = \Sigma_t(x)$ for the NNGP variance.

An infinite ensemble $\bar{f}_t$ equals the mean $m_t$ of the Gaussian process: $\bar{f}_t(x) = m_t(x)$. Note that for $t \to \infty$, the output of the infinite ensemble $\bar{f}_\infty$ on the training inputs $\mathcal{X}$ converges to the targets $\mathcal{Y}$:

$$\bar{f}_\infty(\mathcal{X}) = m_\infty(\mathcal{X}) = \Theta(\mathcal{X}, \mathcal{X})\Theta(\mathcal{X}, \mathcal{X})^{-1}T_\infty\mathcal{Y} = \mathcal{Y}$$

# B    Proof of Theorem 1

In this section, we will go through the steps for the proof of the main theorem of section 4. We first repeat the definition of the GTI-ZSPT and associated discrepency measure $\kappa$

**Definition 1** (Generalisation through invariance ZSPT). *Let $\mathcal{M}|_C$ be a CMDP and let $C_{train}, C_{test} \subset C$ be a set of training and testing contexts that define a ZSPT problem. Additionally, let $\pi^*$ be the optimal policy in $\mathcal{M}|_C$, $S^{\pi^*}_{\mathcal{M}|_C} = \{s \in S | \rho^{\pi^*}_{\mathcal{M}|_C}(s) > 0\}$ denote the set of states with non-zero support under the on-policy distribution $\rho^{\pi^*}_{\mathcal{M}|_C}$ in CMDP $\mathcal{M}|_C$. In the generalisation through invariance ZSPT (GTI-ZSPT), the sets $S^{\pi^*}_{\mathcal{M}|_C}$ and $S^{\pi^*}_{\mathcal{M}|_{C_{train}}}$ admit a symmetric structure:*

$$S^{\pi^*}_{\mathcal{M}|_C} = \{\psi_S(g)s | g \in G, s \in \bar{S}\}$$
$$S^{\pi^*}_{\mathcal{M}|_{C_{train}}} = \{\psi_S(b)s | b \in B, s \in \bar{S}\}, \quad B \leq G$$

*where $\bar{S} \subset S^{\pi^*}_{\mathcal{M}|_{C_{train}}}$ is a proper subset of $S^{\pi^*}_{\mathcal{M}|_{C_{train}}}$ and $G$ is a non-trivial symmetry group (and $B \leq G$ a finite subgroup) that leaves the optimal policy invariant: $\pi^*(s) = \pi^*(\psi_S(g)s), \forall s \in \bar{S}$.*

**Definition 2.** *For the group $G$ and its finite subgroup $B \leq G$ that define the symmetric structure of a GTI-ZSPT (Definition 1), $\kappa$ is a measure of discrepancy between the representations of these groups:*

$$\kappa = \max_{g \in G} \min_{b \in B} ||\psi_S(g) - \psi_S(b)||_{op}$$

*where $|| \cdot ||_{op}$ denotes the operator norm.*

## B.1    Invariance of an ensemble

In order to prove Theorem 1, we first repeat Lemma 6.2 from Gerken and Kessel (2024) that bounds the invariance of an infinitely large ensemble of infinitely wide neural networks trained with full data augmentation on some finite subgroup $B \leq G$:

**Lemma 6.2.** *Let $\pi_\theta : S \to \mathbb{R}$ be an infinitely wide neural network with parameters $\theta$ and with Lipschitz continuous derivatives with respect to the parameters. Furthermore, let $\bar{\pi}_t$ be an infinite ensemble $\bar{\pi}_t(s) = \mathbb{E}_{\theta \sim \mu}[\pi_{\mathcal{L}_t \theta}(s)]$, where the initial weights $\theta$ are sampled from a distribution $\mu$ and the operator $\mathcal{L}_t$ maps $\theta$ to its corresponding value after $t$ steps of gradient descent with respect to a MSE loss function. Define the error $\kappa$ as a measure of discrepancy between representations from the group $G$ and its finite subgroup $B$:*

$$\kappa = \max_{g \in G} \min_{b \in B} ||\psi_S(g) - \psi_S(b)||_{op} \tag{10}$$

*The prediction of an infinite ensemble trained with full data augmentation on $B \leq G$ deviates from invariance by*

$$\left|\bar{\pi}_t(s) - \bar{\pi}_t(\psi_S(g)s)\right| \leq \kappa \, C_\Theta(s), \qquad \forall g \in G \tag{11}$$

*for any time $t$. Here $s \in S$ can by any state and $C_\Theta$ is independent of $g$.*

*Proof.* For completeness we repeat the proof in our own notation in E.2. $\qquad\square$

Next, we prove a lemma that bounds the prediction error between a finite ensemble and an infinite ensemble:

**Lemma 1.** *The difference between the infinite ensemble $\bar{\pi}_t$ and its finite Monte Carlo estimate $\hat{\pi}_t$ of size $N$, is bounded by*

$$|\bar{\pi}_t(s) - \hat{\pi}_t(s)| \leq \frac{1}{\sqrt{N}} C_{\Sigma_t}(s, \epsilon) \tag{12}$$

*with probability at least $1 - \epsilon$. Here $\Sigma_t$ is the variance of the NNGP at time $t$ and $C_{\Sigma_t}(s, \epsilon)$ depends on $\Sigma_t$, the state $s$ and confidence level $\epsilon$.*

*Proof.* We start with Lemma B.4 from Gerken and Kessel (2024) that holds for any Monte-Carlo estimator:

**Lemma B.4.** *The probability that the deep ensemble $\bar{\pi}_t$ and its Monte-Carlo estimate $\hat{\pi}_t$ differ by more than a given threshold $\delta$ is bounded by*

$$\mathbb{P}\big[|\bar{\pi}_t(s) - \hat{\pi}_t(s)| > \delta\big] \leq \sqrt{\frac{2}{\pi}}\frac{\sigma_s}{\delta}\exp\Big(-\frac{\delta^2}{2\sigma_s^2}\Big),$$

*where we have defined*

$$\sigma_s^2 := Var(\hat{\pi}_t)(s) = \frac{\Sigma_t(s)}{N}$$

*where $\Sigma_t(s)$ is the NNGP variance and $N$ is the finite ensemble size.*

*Proof.* For completeness we repeat the proof in our own notation in E.3. $\qquad\square$

We can use this lemma to bound the probability of the deviation between finite and infinite ensemble to be smaller than a threshold $\delta$:

$$\begin{aligned}
\mathbb{P}\big[|\bar{\pi}_t(s) - \hat{\pi}_t(s)| \leq \delta\big] &= 1 - \mathbb{P}\big[|\bar{\pi}_t(s) - \hat{\pi}_t(s)| > \delta\big] \\
&\geq 1 - \epsilon
\end{aligned}$$

where $\epsilon = \sqrt{\frac{2}{\pi}}\frac{\sigma_s}{\delta}\exp\big(-\frac{\delta^2}{2\sigma_s^2}\big)$. Next, we rewrite $\delta$ in terms of a given confidence level $\epsilon$:

$$\begin{aligned}
\epsilon &= \sqrt{\frac{2}{\pi}}\frac{\sigma_s}{\delta}\exp\big(-\frac{\delta^2}{2\sigma_s^2}\big) \\
\frac{2}{\pi\epsilon^2} &= \frac{\delta^2}{\sigma_s^2}\exp\big(\frac{\delta^2}{\sigma_s^2}\big) \\
\frac{\delta^2}{\sigma_s^2} &= W_0\big(\frac{2}{\pi\epsilon^2}\big) \\
\delta &= \sigma_s\sqrt{W_0\big(\frac{2}{\pi\epsilon^2}\big)}
\end{aligned}$$

where $W_0$ is the principal branch of the Lambert $W$ function and the second to last step holds because $\frac{\delta^2}{\sigma_s^2}, \frac{2}{\pi\epsilon^2} \in \mathbb{R}$ and $\frac{2}{\pi\epsilon^2} \geq 0$ for a given probability $\epsilon$. If we know the value for $\epsilon$, $W_0(\frac{2}{\pi\epsilon^2})$ can be solved for numerically. However, in general, the principal branch of the Lambert $W$ function has no closed-form solution, but was upper bounded by [Hoorfar and Hassani (2008)](#)

$$W_0(x) \leq \ln\left(\frac{2x+1}{1+\ln(x+1)}\right)$$

for $x \geq -1/e$. Which means we can upper bound $\delta$ with:

$$\delta \leq \sqrt{\frac{\Sigma_t(s)}{N}}\sqrt{\ln\left(\frac{4+\pi\epsilon^2}{\pi\epsilon^2 + \pi\epsilon^2\ln(2+\pi\epsilon^2) + \pi\epsilon^2\ln(\pi\epsilon^2)}\right)} \tag{13}$$

$$\leq \frac{1}{\sqrt{N}}C_{\Sigma_t}(s,\epsilon) \tag{14}$$

$$\square$$

We can now prove an intermediate lemma that bounds the deviation from the optimal policy for a finite ensemble (rather than an infinite one, as in Lemma 6.2)

**Lemma 2.** *Let the student policy $\hat{\pi}_\infty$ be an ensemble of $N$ infinitely wide neural networks $\pi_\theta$ with Lipschitz continuous derivatives with respect to its parameters, distilled on an on-policy dataset $\mathcal{D} = S_{\mathcal{M}|_{C_{train}}}^{\pi_\beta}$ consisting of all the states encountered by the teacher in $\mathcal{M}|_{C_{train}}$.*

*If the teacher is optimal in the training tasks $C_{train}$ (but arbitrarily bad anywhere else), the deviation from the optimal policy for any test state $s' \in S_{\mathcal{M}|_{C_{test}}}^{\pi^*}$ is bounded with probability at least $1 - \epsilon$, by:*

$$|\pi^*(s') - \hat{\pi}_\infty(s')| \leq \kappa C_\Theta(s') + \frac{1}{\sqrt{N}}C_{\Sigma_\infty}(s',\epsilon)$$

*where $\kappa$ is the measure of discrepancy between subgroup $B \leq G$ and full group $G$ (see definition 2) and $C_\Theta, C_{\Sigma_\infty}$ depend on the state $s' \in S^{\pi^*}_{\mathcal{M}|_{C_{test}}}$, the NTK $\Theta$ (i.e. network architecture), and the dataset $\mathcal{D}$. Additionally, $C_{\Sigma_\infty}$ also depends on the NNGP kernel $\mathcal{K}$ (i.e. network initialisation) and the confidence level $\epsilon$.*

*Proof.* Because we assume the teacher is optimal in the training tasks, our training dataset is actually $\mathcal{D} = S^{\pi_\beta}_{\mathcal{M}|_{C_{train}}} = S^{\pi^*}_{\mathcal{M}|_{C_{train}}}$. Furthermore, by definition of the GTI-ZSPT setting, we have for the states encountered by the optimal policy in $\mathcal{M}|_C$: $S^{\pi^*}_{\mathcal{M}|_C} = \{\psi_S(g)s | g \in G, s \in \bar{S}\}$. Furthermore, we have for the states encountered by the optimal policy in $\mathcal{M}|_{C_{train}}$: $\mathcal{M}|_C$: $S^{\pi^*}_{\mathcal{M}|_{C_{train}}} = \{\psi_S(b)s | b \in B, s \in \bar{S}\}$ for $B \leq G$. This means that for any state $s \in S^{\pi^*}_{\mathcal{M}|_C}$, there exists a symmetry transformation $g^{-1} \in G$ from $s$ to a state $\bar{s} \in \bar{S} \subset \mathcal{D}$ in the training dataset that leaves the policy invariant:

$$\forall s \in S^{\pi^*}_{\mathcal{M}|_C}, \quad \exists g^{-1} \in G, \qquad s.t. \qquad \psi_S(g^{-1})s = \bar{s} \wedge \pi^*(s) = \pi^*(\bar{s}), \quad \text{for some } \bar{s} \in \mathcal{D} \quad (15)$$

Since this holds for any state in $S^{\pi^*}_{\mathcal{M}|_C}$, it also holds for any state in $S^{\pi^*}_{\mathcal{M}|_{C_{test}}} \subset S^{\pi^*}_{\mathcal{M}|_C}$.

Now, Lemma 6.2 holds for any state $s \in S$ and any $g \in G$. So, if we choose $s = \bar{s}$ and $g$ such that $\psi_S(g)\bar{s} = s'$ for a testing state $s' \in S^{\pi^*}_{\mathcal{M}|_{C_{test}}}$, we have

$$\left| \bar{\pi}_t(\bar{s}) - \bar{\pi}_t(\psi_S(g)\bar{s}) \right| \leq \kappa\, C_\Theta(\bar{s})$$
$$\left| \bar{\pi}_t(\bar{s}) - \bar{\pi}_t(s') \right| \leq \kappa\, C_\Theta(s')$$

where we write that $C_\Theta$ is now a function of $s' \in S^{\pi^*}_{\mathcal{M}|_{C_{test}}}$ instead of $\bar{s} \in \mathcal{D}$, which we can do because there exists a one-to-one mapping between the two: $\bar{s} = \psi_S(g^{-1})s'$. The above bound holds for any time $t$. If we choose $t \to \infty$, we have that the infinite ensemble of infinitely wide neural networks $\bar{\pi}_t$ trained on $\mathcal{D}$, will converge to $\bar{\pi}_\infty(\bar{s}) = \pi_\beta(\bar{s}) = \pi^*(\bar{s})$, $\forall \bar{s} \in \mathcal{D}$. Furthermore, due to our choice of $g \in G$, we have that $\bar{\pi}_\infty(\bar{s}) = \pi^*(\bar{s}) = \pi^*(s')$, and the bound becomes

$$\left| \pi^*(s') - \bar{\pi}_\infty(s') \right| \leq \kappa\, C_\Theta(s'), \qquad \forall s' \in S^{\pi^*}_{\mathcal{M}|_{C_{test}}}$$

This bounds the output of the infinite ensemble after training $\bar{\pi}_\infty$, evaluated in a testing state $s' \in S^{\pi^*}_{\mathcal{M}|_{C_{test}}}$, to the optimal policy in that state. We can now combine this bound with our Lemma 1 above to bind the policy of a finite ensemble to the optimal policy in any testing state:

$$\begin{aligned}
|\pi^*(s') - \hat{\pi}_\infty(s')| &= |\pi^*(s') - \bar{\pi}_\infty(s') + \bar{\pi}_\infty(s') - \hat{\pi}_\infty(s')| \\
&\leq |\pi^*(s') - \bar{\pi}_\infty(s')| + |\bar{\pi}_\infty(s') - \hat{\pi}_\infty(s')| \\
&\leq \kappa\, C_\Theta(s') + |\bar{\pi}_\infty(s') - \hat{\pi}_\infty(s')| \\
&\leq \kappa\, C_\Theta(s') + \frac{1}{\sqrt{N}} C_{\Sigma_\infty}(s', \epsilon) \quad \text{with probability } \geq 1 - \epsilon, \quad \forall s' \in S^{\pi^*}_{\mathcal{M}|_{C_{test}}}
\end{aligned}$$

$\square$

### B.2  Performance during testing

We can now use Theorem 3 from Maran et al. (2023) to prove a performance bound for our student policy $\hat{\pi}_\infty(s')$ in the testing CMDP $\mathcal{M}|_{C_{test}}$ in terms of the Wasserstein distance between the student and optimal policy in this testing CMDP:

**Theorem 3.** *Let $\pi^*$ be the optimal policy and $\hat{\pi}_\infty$ be the student policy. If the CMDP is $(L_T, L_R)$-Lipschitz continuous and the optimal and student policies are $L_\pi$-Lipschitz continuous, and we have that $\gamma L_T(1 + L_{\hat{\pi}_\infty}) < 1$, then it holds that:*

$$J^{\pi^*} - J^{\hat{\pi}_\infty} \leq \frac{L_R}{(1-\gamma)(1 - \gamma L_T(1 + L_{\hat{\pi}_\infty}))} \mathbb{E}_{s \sim d^{\pi^*}} \left[ \mathcal{W}(\pi^*(\cdot|s), \hat{\pi}_\infty(\cdot|s)) \right]$$

*where $d^{\pi^*}(s) = (1-\gamma) \sum_{t=0}^\infty \gamma^t \mathbb{P}(s_t = s | \pi^*, p_0)$ is the $\gamma$-discounted visitation distribution and $\gamma$ the the discount factor.*

*Proof.* For completeness we repeat the proof in our notation in Appendix E.1. □

With this, we can finally prove the main theorem:

**Theorem 1.** *Consider policy distillation for a deterministic, scalar teacher policy $\pi_\beta : S \to \mathbb{R}$ (Equation (1) in Section 2.1) in a $L_T, L_R$-Lipschitz continuous CMDP in the GTI-ZSPT setting. Let the student policy $\hat{\pi}_\infty$ be an ensemble of $N$ infinitely wide neural networks $\pi_\theta : S \to \mathbb{R}$ with Lipschitz continuous derivatives with respect to its parameters, distilled on an on-policy dataset $\mathcal{D} = S_{\mathcal{M}|_{C_{train}}}^{\pi_\beta} = \{\psi_S(b)s | b \in B, s \in \bar{S}\}$ consisting of all the states in the training contexts encountered by the teacher in the GTI-ZSPT setting. Furthermore, let the student policy be $L_{\hat{\pi}_\infty}$-Lipschitz continuous and assume $\gamma L_T(1 + L_{\hat{\pi}_\infty}) < 1$.*

*If the teacher is optimal in the training tasks $C_{train}$ (but arbitrarily bad anywhere else), the performance of the student in the testing CMDP $\mathcal{M}|_{C_{test}}$ is bounded with probability at least $1 - \epsilon$, by:*

$$J^{\pi^*} - J^{\hat{\pi}_\infty} \leq \frac{L_R}{(1-\gamma)(1-\gamma L_T(1+L_{\hat{\pi}_\infty}))}\left(\kappa \bar{C}_\Theta + \frac{1}{\sqrt{N}}\bar{C}_{\Sigma_\infty}(\epsilon)\right) \tag{5}$$

*where $\kappa$ is the measure of discrepancy between subgroup $B \leq G$ and full group $G$ (see definition 2) and $\bar{C}_\Theta, \bar{C}_{\Sigma_\infty}$ are constants that depend on the $\gamma$-discounted visitation distribution of the optimal policy in $\mathcal{M}|_{C_{test}}$, the network architecture, and the dataset $\mathcal{D}$. Additionally, $\bar{C}_{\Sigma_\infty}$ also depends on the network initialisation and the confidence level $\epsilon$.*

*Proof.* We have for deterministic polices that the Wasserstein distance reduces to

$$\mathcal{W}(\pi^*(\cdot|s), \hat{\pi}_\infty(\cdot|s)) = |\pi^*(s) - \hat{\pi}_\infty(s)|$$

So, if we invoke Theorem 3 from Maran et al. (2023) on the testing CMDP $\mathcal{M}|_{C_{test}}$, we can use Lemma 2 to show

$$
\begin{aligned}
J^{\pi^*} - J^{\hat{\pi}_\infty} &\leq \frac{L_R}{(1-\gamma)(1-\gamma L_T(1+L_{\hat{\pi}_\infty}))}\mathbb{E}_{s\sim d^{\pi^*}}[\mathcal{W}(\pi^*(\cdot|s), \hat{\pi}_\infty(\cdot|s))] \\
&\leq \frac{L_R}{(1-\gamma)(1-\gamma L_T(1+L_{\hat{\pi}_\infty}))}\mathbb{E}_{s\sim d^{\pi^*}}[|\pi^*(s) - \hat{\pi}_\infty(s)|] \\
&\leq \frac{L_R}{(1-\gamma)(1-\gamma L_T(1+L_{\hat{\pi}_\infty}))}\left(\kappa \bar{C}_\Theta + \frac{1}{\sqrt{N}}\bar{C}_{\Sigma_\infty}(\epsilon)\right)
\end{aligned}
$$

where $\bar{C}_\Theta = \mathbb{E}_{s\sim d^{\pi^*}}[C_\Theta(s)]$ depends on the NTK $\Theta$ (i.e. network architecture) and $C_{\Sigma_\infty}(\epsilon) = \mathbb{E}_{s\sim d^{\pi^*}}[C_{\Sigma_\infty}(s, \epsilon)]$ depends on the NNGP kernel $\mathcal{K}$ (i.e. network initialisation). □

## C  Experimental details

The code for all the experiments in the main text can be found at https://github.com/MWeltevrede/distillation-after-training.

### C.1  'Reacher with rotational symmetry' CMDP

In the 'Reacher with rotational symmetry' CMDP from Figure 1, the state $s = (x_s, y_s, x_e, y_e, x_h, y_h)$ consists of the 2D Euclidean coordinates of the shoulder $(x_s, y_s)$, elbow $(x_e, y_e)$ and hand $(x_h, y_h)$ centred around the target location, and the continuous 2D action space consists of the torque to rotate the shoulder and elbow joints. The episode terminates and the agent receives a reward of 1 if the hand of the robot arm is within a small area around the target location. Elsewhere, the reward function equals $\frac{1 - 0.5d_{target}}{0.5T} \delta_{d_{target} < d_{min}}$, where $d_{target}$ is the distance between the target location and hand, $T = 200$ is the maximum number of steps before timeout, and $\delta_{d_{target} < d_{min}}$ is 1 only when the current $d_{target}$ is smaller than the minimal distance $d_{min}$ to target achieved in that episode, and 0 otherwise. In the experiments from Section 5.1, policies are distilled on datasets collected by rolling out trajectories from a teacher agent in a fixed set of training contexts. Ensembles are created by independently distilling $N$ policies (with different seeds) and afterwards evaluating by averaging over the output of the $N$ polices.

#### C.1.1  Satisfying the assumptions for the GTI-ZSPT setting

The 'Reacher with rotational symmetry' CMDP from Figure 1 satisfies the symmetric structure assumed in the GTI-ZSPT setting. To illustrate this, we could define the states encountered by the optimal policy in context 1 in Figure 1 as the subset of states $\bar{S}$ in the GTI-ZSPT definition. With that definition we can see that subgroup $B = C_4$ would generate all the states in $S^{\pi^*}_{\mathcal{M}|_{C_{train}}}$, and full group $G = SO(2)$ would generate all the states in $S^{\pi^*}_{\mathcal{M}|_C}$.

For the states $s = (x_s, y_s, x_e, y_e, x_h, y_h)$, the representation $\psi_S(\alpha)$ for a rotation with angle $\alpha$ is the block diagonal matrix:

$$
\psi_S(\alpha) = \begin{bmatrix}
\cos\alpha & -\sin\alpha & 0 & 0 & 0 & 0 \\
\sin\alpha & \cos\alpha & 0 & 0 & 0 & 0 \\
0 & 0 & \cos\alpha & -\sin\alpha & 0 & 0 \\
0 & 0 & \sin\alpha & \cos\alpha & 0 & 0 \\
0 & 0 & 0 & 0 & \cos\alpha & -\sin\alpha \\
0 & 0 & 0 & 0 & \sin\alpha & \cos\alpha
\end{bmatrix}
$$

which is orthogonal.

Additionally, the CMDP is $L_T$-Lipschitz continuous since there are no collisions causing non-smooth transitions. Moreover, with a proper choice of reward function (for example, $R = \frac{1}{d_{target}}$), it is also $L_R$-Lipschitz continuous. Note that for our experiments, we choose a non smooth reward function since it helped with training the teacher.

#### C.1.2  Figure 1 & Table 1

In the setting from Figure 1 and Table 1, the contexts only differ in the location of the shoulder. The robot arm pose always starts at a $45°$ degree angle for the shoulder joint (counter-clockwise with respect to an axis drawn from the shoulder to the target), and a $90°$ degree angle for the elbow joint (clockwise with respect to an axis drawn from the shoulder to elbow). In the testing distribution, the shoulder can be located anywhere along a circle around the target location, but the starting pose is always the same.

For the training contexts, the shoulder is located at different, evenly spaced, intervals around the $360°$ circle. In the bottom half of Table 1, we train on three different data sets denoted by the corresponding subgroups of $SO(2)$ that generate the training contexts: $C_2, C_4$ and $C_8$. For each of the datasets, we always context 1 in Figure 1 as the base context, and apply various rotations to generate the other training contexts. The $C_2$ set consists of context 1 together with the subgroup of $180°$ rotations (the $0°$ and $180°$ rotations, resulting in context 1 and context 3 in Figure 1). The set $C_4$ consists of the $90°$ rotations and the corresponding four training contexts are depicted in Figure 1. Lastly, the $C_8$ set

consists of the $45°$ rotations, half of which are the $90°$ rotations from Figure 1, and the other half are the $45°$ rotations in between those. Note that for the results of varying ensemble size in the top half of Table 1, we used the $C_4$ dataset.

The teacher policy is a handcrafted policy ($a = (-2, 2)$ for 12 steps and $a = (2, 2)$ afterwards) that is optimal for the starting pose considered in this setting. The neural network consists of three fully connected hidden layers of size $[64, 64, 32]$ with ReLU activation functions. The policy is distilled with the MSE loss in (6) (which is the same loss as (1) but for vector-valued actions instead of scalar). The exact hyperparameters can be found in Table 4.

Table 4: Hyper-parameters used for the 'Reacher with rotational symmetry' CMDP experiments

| 'Reacher with rotational symmetry' | |
| --- | --- |
| **Hyper-parameter** | **Value** |
| Epochs | 500 |
| Batch size | 6 |
| Learning rate | $1 \times 10^{-4}$ |

### C.1.3   Figure 2 & Table 2

In the setting from Figure 2 and Table 2, the contexts not only differ in the shoulder location (as described in the subsection above), but also in the starting pose of the robot arm. For testing, a random shoulder location and starting pose are sampled for each episode.

There are four training contexts in this setting, whose shoulder location correspond to the $C_4$ dataset described above, but whose initial arm poses are sampled randomly by sampling two angles between 0 and 360 degrees for the shoulder and elbow joint. Each seed has its own set of random training poses (but the same shoulder location). The dataset created from these four training contexts is referred to as the *Training Contexts* dataset in Table 2. The *Training Contexts* + $C_4$ dataset essentially consists of 16 training contexts, four of which are the ones from Training Contexts, and the other 12 are the random poses from the Training Contexts contexts, duplicated for each of the other shoulder locations. Figure 3 illustrates this for an example set of four Training Contexts. The *Training Contexts* + *Random* dataset is the same as the Training Contexts + $C_4$ set, except that instead of duplicating the random poses from the four Training Contexts, 12 new random poses are sampled.

For this more complicated version of the 'Reacher with rotational symmetry' CMDP, it is much more convoluted to handcraft an optimal policy. Instead, we train an soft actor-critic agent (SAC Haarnoja et al., 2018) with the Stable-Baselines3 (Raffin et al., 2021) implementation on the full context distribution, to get close to an optimal policy for any context. The network for the SAC teacher consists of two fully connected hidden layers of size $[400, 300]$ with ReLU activation functions. The other hyperparameters for the SAC agent can be found in 5. Note that for the results in Table 2, we evaluate single distilled policies ($N = 1$) and we use the same network and hyperparameters for distillation as the experiments for Table 1.

Table 5: Hyper-parameters used for the SAC teacher agent in the 'Reacher with rotational symmetry' CMDP

| SAC Teacher | |
|---|---|
| **Hyper-parameter** | **Value** |
| Total timesteps | 500 000 |
| Buffer size | 300 000 |
| Batch size | 256 |
| Discount factor $\gamma$ | 0.99 |
| Gradient steps | 64 |
| Train frequency (steps) | 64 |
| Target update interval (steps) | 1 |
| Target soft update coefficient $\tau$ | 0.02 |
| Warmup phase | 10 000 |
| Share feature extractor | False |
| Target entropy | auto |
| Entropy coeff | auto |
| Use State Dependent Exploration (gSDE) | True |
| **Adam** | |
| Learning rate | $5 \times 10^{-4}$ |

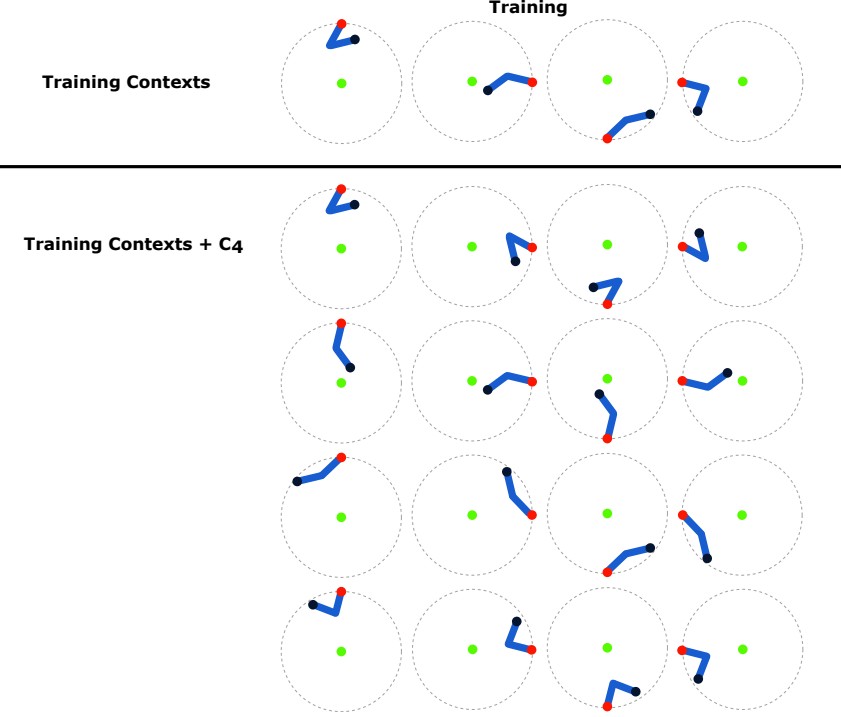

Figure 3: The Training Contexts and Training Contexts + $C_4$ context sets in the 'Reacher with rotational symmetry' reacher CMDP with varying shoulder location (red) *and* robot arm pose (blue), see Figure 1 for details.

## C.2  Four Rooms

In the Four Rooms environment (Figure 4), an agent (red triangle) starts in a random location and facing a random direction, and has to move to the goal location (green square) whilst navigating the doorways connecting the four rooms. We modify the original Minigrid implementation a little bit, by reducing the action space from the default seven (turn left, turn right, move forward, pick up an object,

drop an object, toggle/activate an object, end episode) to only the first three (turn left, turn right, move forward). Moreover, we use a reward function that gives a reward of 1 when the goal is reached and zero elsewhere, which differs from slightly from the default one that gives $1 - 0.9 * \left(\frac{\text{step count}}{\text{max steps}}\right)$ for reaching the goal. Lastly, our implementation of the Four Rooms environment allows for more control over the context dimensions, allowing for the construction of distinct training and testing sets. Our version of the Four Room environment can be found at `<redacted for review>`.

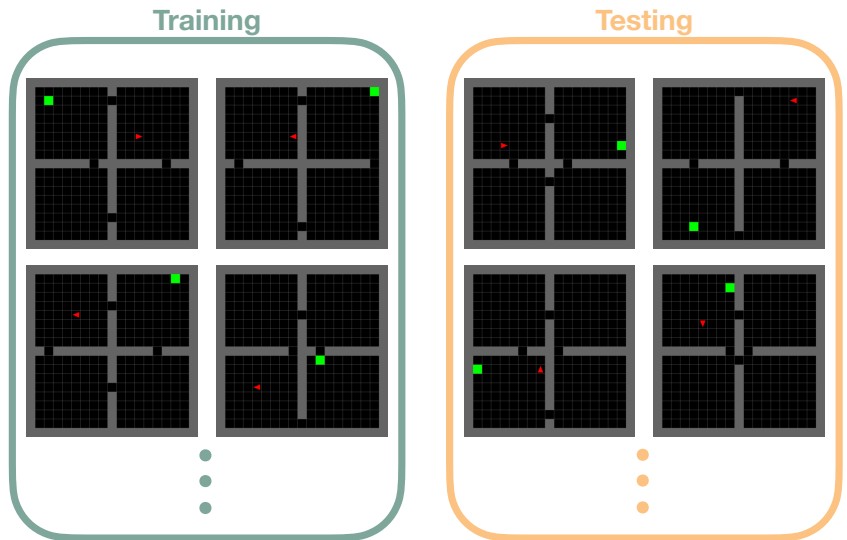

Figure 4: Example of Four Rooms training and testing contexts.

In this environment, the contexts differ in the topology of the doorways connecting the four rooms, the initial location of the agent, the initial direction the agent is facing, and the goal location. The teacher is the PPO+Explore-Go agent from Weltevrede et al. (2025), that is trained on 200 training contexts. We used separate validation and testing sets consisting of unseen contexts of size 40 and 200 respectively. The validation set was used for algorithm development and hyperparameter tuning, and the test set was only used as a final evaluation (and is reported in Table 3). Since the optimal policy in Four Rooms is deterministic, we also evaluate performance of our policies deterministically by always taking the action with maximum probability in a given state.

The *Teacher* dataset used for distillation and behaviour cloning, simply consists of the states encountered by rolling out the stochastic teacher policy in the 200 training contexts until the desired dataset size was reached. We use a dataset size of 500.000, since this was the replay buffer size of the DQN agent from Weltevrede et al. (2025). The *Explore-Go* dataset mimics what a rollout buffer would look like for the PPO+Explore-Go teacher. It is created by running a pure exploration agent (trained as part of the PPO+Explore-Go teacher) for $k$ steps at the beginning of each episode, and afterwards rolling out the stochastic teacher policy until termination of the episode. The number of pure exploration steps $k$ is sampled uniformly from a range $[0, K)$, where $K = 50$ (the same as was used in Weltevrede et al. (2025)). The pure exploration experience is not added to the dataset, only the states encountered by the teacher. The Explore-Go dataset also has size 500.000, but requires additional interactions with the environment (on average $\frac{K}{2} = 25$ steps per episode) that are not added to the dataset. Since the average episode length of the teacher is of similar size, the Explore-Go dataset requires roughly twice the dataset size of additional environment steps to create. Lastly, the *mixed* dataset is a 50/50 mixture of a Teacher dataset of size 250.000, and a dataset created by rolling out the pure exploration policy for 250.000 steps. We generate 20 PPO+Explore-Go teachers, and generate one dataset of each type per teacher (for a total of 20 datasets).

An important thing to note, is that although the states for the distillation and behaviour cloning experiments are the same, the learning targets are not. This has the biggest effect on the Mixed dataset, where the learning targets for behaviour cloning are the actions that were taken to create the dataset, and for distillation, the targets are the PPO+Explore-Go teacher's probabilities in the given state (independent of what action was taken in that state during the creation of the dataset).

The Four Room experiments were executed on a computer with an NVIDIA RTX 3070 GPU, Intel Core i7 12700 CPU and 32 GB of memory. Training of the teacher (PPO agent) would take approximately 2 hours, and a single distillation run would take approximately 10 minutes. The code for our experiments can be found at `<redacted for review>`.

### C.2.1 Implementation details

For the distillation experiments (top of Table 3), we used the same architecture as the teacher in Weltevrede et al. (2025). We distil the stochastic teacher policy by regressing on the probabilities (as in Equation 7). We found this to work significantly better than alternative loss functions, since the normalised range for the targets helps the averaging in the ensemble. We tune the distillation hyperparameters by performing a grid search over the following values

- **Learning rate**: $\{1 \times 10^{-4}, 1 \times 10^{-3}, 1 \times 10^{-2}\}$
- **Batch Size**: $\{64, 256, 512, 1024, 2048\}$
- **Epochs**: $\{10, 20, 30, 40, 50, 60, 70, 80, 90, 100\}$

We performed the tuning for a single teacher seed by, for each dataset type, splitting the dataset into a training set (sampled from the first 150 training contexts) and validation set (sampled from the other 50 training contexts), and choosing the combination of hyperparameters that minimised the distillation loss on the validation set. The final results are distilled on the full datasets and evaluated in the testing contexts. The final hyperparameters can be found in Table 6.

Table 6: Hyperparameters used for policy distillation in the Four Rooms environment.

| Four Rooms Distillation | |
|---|---|
| **Hyper-parameter** | **Value** |
| **Teacher** | |
| Epochs | 100 |
| Batch size | 64 |
| Learning rate | $1 \times 10^{-4}$ |
| **Explore-Go** | |
| Epochs | 50 |
| Batch size | 512 |
| Learning rate | $1 \times 10^{-3}$ |
| **Mixed** | |
| Epochs | 50 |
| Batch size | 256 |
| Learning rate | $1 \times 10^{-3}$ |

For the behaviour cloning experiments (bottom of Table 3), we used the same architecture as for the distillation experiments. We trained using the logarithmic BC loss for stochastic policies (Equation (8)). We also tuned the behaviour cloning in the same way as the distillation policies above, but with a smaller range for the epochs: $\{1, 2, 3, 4, 5, 6, 7, 8, 9, 10\}$, since we found it would overfit much sooner than the distillation experiments. The final hyperparameters can be found in Table 7.

Table 7: Hyperparameters used for behaviour cloning in the Four Rooms environment.

| Four Rooms Behaviour Cloning | |
|---|---|
| **Hyper-parameter** | **Value** |
| **Teacher** | |
| Epochs | 1 |
| Batch size | 64 |
| Learning rate | $1 \times 10^{-3}$ |
| **Explore-Go** | |
| Epochs | 1 |
| Batch size | 64 |
| Learning rate | $1 \times 10^{-3}$ |
| **Mixed** | |
| Epochs | 2 |
| Batch size | 256 |
| Learning rate | $1 \times 10^{-3}$ |

# D Additional experiments

## D.1 Measure of invariance

In order to identify invariance as a key factor for the increased performance in the 'Reacher with rotational symmetry' environment from Table 1, we will measure how invariant the policy becomes when increasing ensemble or subgroup size. As a measure of invariance we use the variance of the network's output across the group orbit (Kvinge et al., 2022) (for a completely invariant network, this should be zero). In practical terms, we evaluate each policy on all (360) integer degree rotations of the starting state (the orbit) and compute the total variation (trace of the covariance matrix) over the produced outputs. In Figure 5, we see that for both increased ensemble and subgroup size, as the test performance increases, so does the measure of invariance decrease.

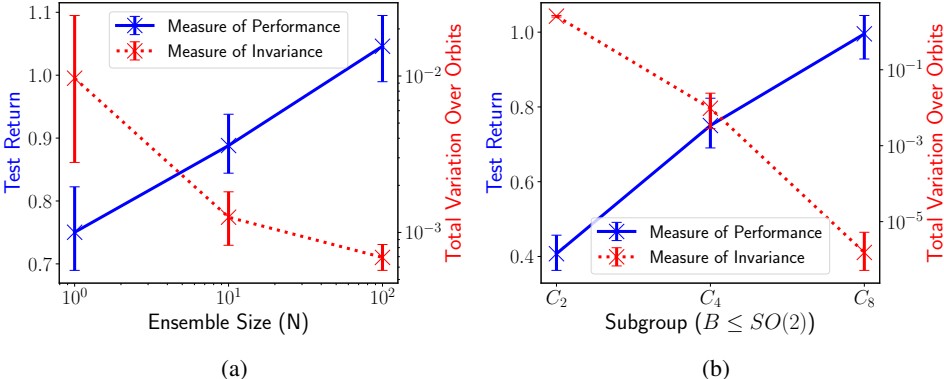

(a)                                        (b)

Figure 5: Test return (left axis) compared with the total variation (trace of the covariance matrix) over orbits of the $SO(2)$ group of rotations (right axis) for (a) different ensemble sizes and (b) subgroups $B \leq SO(2)$. The total variation is a measure of how invariant the agent has become with respect to rotations, zero total variation would correspond to perfect invariance. Shown are the mean and 95% confidence intervals over 20 seeds.

## D.2 $\frac{1}{\sqrt{N}}$ generalisation bound

In this section, we investigate how well our results fit the $a + \frac{b}{\sqrt{N}}$ form of the generalisation bound from Theory 1. We evaluate different ensemble sizes on the 'Reacher with rotational symmetry' environment from Figure 1, trained on subgroup $B = C_4$ in Table 8. This table includes the results from Table 1, as well as additional results for ensemble sizes $N = 1000$ and $N = 10.000$. In Table 9, we repeat the results for different ensemble sizes distilled on the Explore-Go dataset in the Four Rooms environment from Table 3, with the addition of ensemble size $N = 100$.

In Figure 6, we compare the difference to optimal performance $J^{\pi^*} - J^{\hat{\pi}\infty}$ with ensemble size $N$, and plot the best fit to the $a + \frac{b}{\sqrt{N}}$ relation as predicted by our theory. The exact optimal performance $J^{\pi^*}$ is not necessarily known, but we estimate it by taking the average train performance instead (which seems to have converged on both environments). The best $a + \frac{b}{\sqrt{N}}$ fit is obtained by computing the optimal linear fit between $y = J^{\pi^*} - J^{\hat{\pi}\infty}$ and $x' = \frac{1}{\sqrt{N}}$ (and then transforming the solution back to $x = N$).

The results in Figure 6 seem to follow the $a + \frac{b}{\sqrt{N}}$ upper bound to some extend. The upper bound is likely not very tight, but the results seem to indicate the bound is also not vacuous.

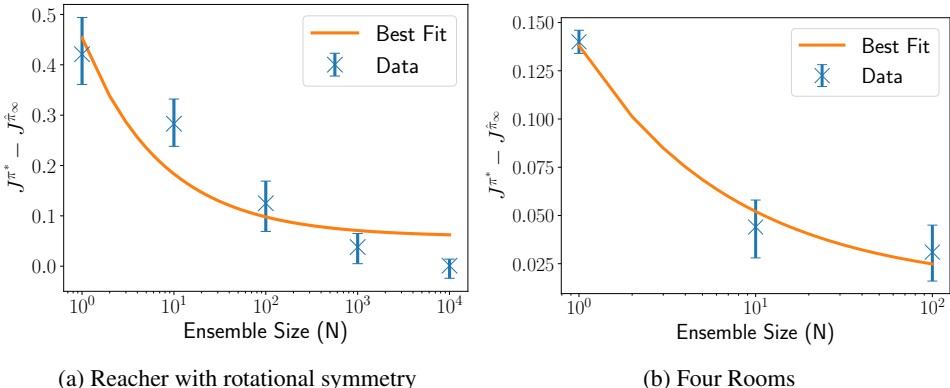

(a) Reacher with rotational symmetry      (b) Four Rooms

Figure 6: Difference to optimal performance as a function of ensemble size $N$ in the testing contexts for the (a) Reacher with rotational symmetry and (b) Four Rooms environments. Shown are the mean and 95% confidence intervals over 20 seeds from (a) Table 8 and (b) Table 9, together with the best possible fit for the relation $a + \frac{b}{\sqrt{N}}$ as predicted by our theory.

Table 8: Performance of distilled policies in the Illustrative CMDP from Figure 1 for different ensemble sizes $N$ (trained under subgroup $B = C_4$). Shown are the mean and standard deviation for 20 seeds, and in bold are the best returns including those with overlapping 95% confidence intervals.

| Ensemble Size $N$: | N=1 | N=10 | N=100 | N=1000 | N=10.000 |
|---|---|---|---|---|---|
| Train Performance | **1.17 ± 0.004** | **1.17 ± 0.004** | **1.17 ± 0.003** | **1.17 ± 0.003** | **1.17 ± 0.002** |
| Test Performance | 0.75 ± 0.147 | 0.89 ± 0.107 | 1.05 ± 0.117 | 1.13 ± 0.072 | **1.17 ± 0.038** |

Table 9: Performance of an ensemble (of size $N$) of policy distillation or behaviour cloning policies on the Explore-Go dataset in the Four Rooms environment. Shown are mean and standard deviation over 20 seeds, and in bold are the best returns including those with overlapping 95% confidence intervals.

| Ensemble Size $N$: | N=1 | N=10 | N=100 |
|---|---|---|---|
| Train Performance | **0.92 ± 0.020** | **0.92 ± 0.019** | **0.92 ± 0.021** |
| Test Performance | 0.78 ± 0.041 | **0.88 ± 0.036** | **0.89 ± 0.033** |

# E Repeated proofs

## E.1 Theorem 3 from Maran et al. (2023)

Here we repeat the proof for Theorem 3 in Maran et al. (2023). We mostly change the notation to be consistent with our paper. Note that in Maran et al. (2023) they consider MDPs rather than CMDPs, but since any CMDP (including the testing CMDP $\mathcal{M}|_{C_{test}}$ in a ZSPT setting) is just a special instance of an MDP, the theorem readily applies.

We first introduce a number of definitions and assumptions used in our derivations

**Definition 3.** *We will call a function $f$ $L$-Lipschitz continuous if for two metric sets $(X, d_x)$, $(Y, d_y)$, where $d_x, d_y$ are distance metrics, we have*

$$\forall x_1, x_2 \in X, \quad d_y(f(x_1), f(x_2)) \leq L d_x(x_1, x_2).$$

**Definition 4.** *We define $\|f\|_L$ to be the Lipschitz semi-norm of $f$, with*

$$\|f\|_L = \sup_{x_1, x_2 \in X, x_1 \neq x_2} \frac{d_y(f(x_1), f(x_2))}{d_x(x_1, x_2)}.$$

**Definition 5.** *We introduce the Wasserstein distance between probability distributions $p$ and $q$ as*

$$\mathcal{W}(p, q) = \sup_{\|f\|_L \leq 1} \left| \int f dp - \int f dq \right|.$$

**Assumption 1.** We assume an $(L_T, L_R)$-Lipschitz continuous MDP with a metric state and action space and associated distances $d_s (\equiv d_S)$ and $d_a (\equiv d_A)$, for which we have

$$\mathcal{W}(T(\cdot|s, a), T(\cdot|\hat{s}, \hat{a})) \leq L_T(d_S(s, \hat{s}) + d_A(a, \hat{a})), \qquad \forall (s, a), (\hat{s}, \hat{a}) \in S \times A$$
$$|R(s, a) - R(\hat{s}, \hat{a})| \leq L_R(d_S(s, \hat{s}) + d_A(a, \hat{a})), \qquad \forall (s, a), (\hat{s}, \hat{a}) \in S \times A.$$

**Assumption 2.** We assume $L_\pi$-Lipschitz continuous policies, which satisfy

$$\mathcal{W}(\pi(\cdot|s), \pi(\cdot|s')) \leq L_\pi d_S(s, s') \quad \forall s, \hat{s} \in S.$$

Note that this definition subsumes deterministic policies.

With this, we can state the theorem in question (Maran et al., 2023)

**Theorem 3.** *Let $\pi^*$ be the optimal policy and $\hat{\pi}_\infty$ be the student policy. If the CMDP is $(L_T, L_R)$-Lipschitz continuous and the optimal and student policies are $L_\pi$-Lipschitz continuous, and we have that $\gamma L_T(1 + L_{\hat{\pi}_\infty}) < 1$, then it holds that:*

$$J^{\pi^*} - J^{\hat{\pi}_\infty} \leq \frac{L_R}{(1 - \gamma)(1 - \gamma L_T(1 + L_{\hat{\pi}_\infty}))} \mathbb{E}_{s \sim d^{\pi^*}} [\mathcal{W}(\pi^*(\cdot|s), \hat{\pi}_\infty(\cdot|s))]$$

*where $d^{\pi^*}(s) = (1 - \gamma) \sum_{t=0}^\infty \gamma^t \mathbb{P}(s_t = s|\pi^*, p_0)$ is the $\gamma$-discounted visitation distribution and $\gamma$ the the discount factor.*

*Proof.* From assumptions 1 and 2, it follows that if $\gamma L_p(1 + L_\pi) < 1$ the value function $Q^\pi$ associated with $\pi$ is $L_{Q^\pi}$-Lipschitz continuous (Rachelson and Lagoudakis, 2010) with

$$L_{Q^\pi} \leq \frac{L_R}{1 - \gamma L_T(1 + L_\pi)}$$

The first part of our proof derives performance differences under Lipschitz value functions. We begin with the performance difference Theorem (Kakade and Langford, 2002) stating

$$J^{\pi_1} - J^{\pi_2} = \frac{1}{1 - \gamma} \mathbb{E}_{s \sim d^{\pi_1}} \left[ \mathbb{E}_{a \sim \pi_1(\cdot|s)} [Q^{\pi_2}(s, a) - V^{\pi_2}(s)] \right]$$

where $V^{\pi_2}(s)$ is the state value function. Focusing on the inner expectation we have

$$
\begin{aligned}
\mathbb{E}_{a\sim\pi_1(\cdot|s)}[Q^{\pi_2}(s,a) - V^{\pi_2}(s)] &= \int_A (Q^{\pi_2}(s,a) - V^{\pi_2}(s))\pi_1(da|s) \\
&= \int_A Q^{\pi_2}(s,a)\pi_1(da|s) - V^{\pi_2}(s) \\
&= \int_A Q^{\pi_2}(s,a)(\pi_1(da|s) - \pi_2(da|s)).
\end{aligned}
$$

Now, let $L_s = \|Q^{\pi_2}(s,\cdot)\|_L$ be the Lipschitz semi-norm of $Q^{\pi_2}(s,a)$ w.r.t. $a$ and define $g_s(a) = Q^{\pi_2}(s,a)/L_s$ with the property $\|g_s\|_L = 1$. This yields

$$
\begin{aligned}
\mathbb{E}_{a\sim\pi_1(\cdot|s)}[Q^{\pi_2}(s,a) - V^{\pi_2}(s)] &= \int_A Q^{\pi_2}(s,a)(\pi_1(da|s) - \pi_2(da|s)) \\
&= \int_A g_s(a)L_s(\pi_1(da|s) - \pi_2(da|s)) \\
&= L_s \int_A g_s(a)(\pi_1(da|s) - \pi_2(da|s))
\end{aligned}
$$

By definition of the Wasserstein distance

$$
\mathcal{W}(\pi_1(\cdot|s), \pi_2(\cdot|s)) = \sup_{\|g\|_L \leq 1} \left| \int_A g(a)(\pi_1(da|s) - \pi_2(da|s)) \right|,
$$

such that we have

$$
\begin{aligned}
\left| \mathbb{E}_{a\sim\pi_1(\cdot|s)}[Q^{\pi_2}(s,a) - V^{\pi_2}(s)] \right| &= \left| L_s \int_A g_s(a)(\pi_1(da|s) - \pi_2(da|s)) \right| \\
&\leq L_s \sup_{\|g\|_L \leq 1} \left| \int_A g(a)(\pi_1(da|s) - \pi_2(da|s)) \right| \\
&= L_s \mathcal{W}(\pi_1(\cdot|s), \pi_2(\cdot|s)).
\end{aligned}
$$

Now, we recall $L_s = \|Q^{\pi_2}(s,\cdot)\|_L$ and by our assumptions $Q^{\pi_2}$ is $L_{Q^{\pi_2}}$-Lipschitz continuous such that

$$
\begin{aligned}
L_s &\leq \sup_{s\in S} \|Q^{\pi_2}(s,\cdot)\|_L \\
&\leq \|Q^{\pi_2}\|_L \\
&\leq L_{Q^{\pi_2}}.
\end{aligned}
$$

Putting these results together, we can obtain

$$
J^{\pi_1} - J^{\pi_2} \leq \frac{L_{Q^{\pi_2}}}{1-\gamma} \mathbb{E}_{s\sim d^{\pi_1}}[\mathcal{W}(\pi_1(\cdot|s), \pi_2(\cdot|s))]
$$

After setting $\pi_1 = \pi^*$ and $\pi_2 = \hat{\pi}_\infty$ and using that $L_{Q^{\hat{\pi}_\infty}} \leq \frac{L_R}{1-\gamma L_T(1+L_{\hat{\pi}_\infty})}$, we have

$$
J^{\pi^*} - J^{\hat{\pi}_\infty} \leq \frac{L_R}{(1-\gamma)(1-\gamma L_T(1+L_{\hat{\pi}_\infty}))} \mathbb{E}_{s\sim d^{\pi^*}}[\mathcal{W}(\pi^*(\cdot|s), \hat{\pi}_\infty(\cdot|s))]
$$

$\square$

### E.2 Lemma 6.2 from Gerken and Kessel (2024)

Here we repeat the proof for Lemma 6.2 in Gerken and Kessel (2024) in the notation used in this paper.

**Lemma 6.2.** *Let $\pi_\theta : S \to \mathbb{R}$ be an infinitely wide neural network with parameters $\theta$ and with Lipschitz continuous derivatives with respect to the parameters. Furthermore, let $\bar{\pi}_t$ be an infinite ensemble $\bar{\pi}_t(s) = \mathbb{E}_{\theta\sim\mu}[\pi_{\mathcal{L}_t\theta}(s)]$, where the initial weights $\theta$ are sampled from a distribution $\mu$ and the operator $\mathcal{L}_t$ maps $\theta$ to its corresponding value after $t$ steps of gradient descent with respect to a*

*MSE loss function. Define the error $\kappa$ as a measure of discrepancy between representations from the group $G$ and its finite subgroup $B$:*

$$\kappa = \max_{g \in G} \min_{b \in B} ||\psi_S(g) - \psi_S(b)||_{op} \qquad (10)$$

*The prediction of an infinite ensemble trained with full data augmentation on $B \leq G$ deviates from invariance by*

$$\left| \bar{\pi}_t(s) - \bar{\pi}_t(\psi_S(g)s) \right| \leq \kappa \, C_\Theta(s), \qquad \forall g \in G \qquad (11)$$

*for any time t. Here $s \in S$ can by any state and $C_\Theta$ is independent of g.*

*Proof.* Lets denote a set of states with $\mathcal{D} = \{s_i\}_{i=1}^n$ and a training dataset $\mathcal{T} = \{(s_i, y_i) | \forall s_i \in \mathcal{D}, y_i \in \mathcal{Y}\}$ where $y_i \in \mathcal{Y}$ indicates the target for sample $s_i$ (for example, $y_i = \pi_\beta(s_i)$ for distillation with respect to a teacher $\pi_\beta$). Using the definition of the measure of discrepancy $\kappa$ and the property of full data augmentation with a finite group from (9), we can write for any training sample $(s_j, y_j) \in \mathcal{T}_B = (\mathcal{D}_B, \mathcal{Y}_B) = \{(\psi_S(b)s, y) | \forall (s, y) \in \mathcal{T}, b \in B\}$ and any $g \in G$ and $b \in B$:

$$||\psi_S(g)s_j - s_{\mathrm{p}_b(j)}|| = ||\psi_S(g)s_j - \psi_S(b)s_j|| \leq ||\psi_S(g) - \psi_S(b)||_{op} ||s_j|| < \kappa ||s_j|| \, .$$

Additionally, we can use the definition of the mean of the NNGP $m_t$ to write for any $s \in S$

$$|\bar{\pi}_t(s) - \bar{\pi}_t(\psi_S(g)s)| = |m_t(s) - m_t(\psi_S(g)s)|$$
$$= |(\Theta(s, \mathcal{D}_B) - \Theta(\psi_S(g)s, \mathcal{D}_B))\Theta^{-1}(\mathbb{I} - \exp(-\eta\Theta t))\mathcal{Y}_B| \, .$$

We can use Lemma 5.2 from Gerken and Kessel (2024), made specific for scalar- and vector-valued functions:

**Lemma 5.2.** *For scalar- and vector-valued functions, data augmentation implies that the permutation $\Pi_g$ commutes with any matrix-valued analytical function F involving the NNGP kernel $\mathcal{K}$, the NTK $\Theta$ and their inverses:*

$$\Pi(g)F(\Theta, \Theta^{-1}, \mathcal{K}, \mathcal{K}^{-1}) = F(\Theta, \Theta^{-1}, \mathcal{K}, \mathcal{K}^{-1})\Pi(g)$$

*where $\Pi(g)$ denotes the permutation matrix applying the permutation $\mathrm{p}_g$ associated with g to each training point.*

to show that:

$$\Theta(s, \mathcal{D}_B)\Theta^{-1}(\mathbb{I} - \exp(-\eta\Theta t))\mathcal{Y}_B = \Theta(s, s_i)\Theta_{ij}^{-1}(\mathbb{I} - \exp(-\eta\Theta t))_{jk}y_k$$
$$= \Theta(s, s_i)\Theta_{ij}^{-1}(\mathbb{I} - \exp(-\eta\Theta t))_{jk}y_{\mathrm{p}_b(k)}$$
$$= \Theta(s, s_{\mathrm{p}_b^{-1}(i)})\Theta_{ij}^{-1}(\mathbb{I} - \exp(-\eta\Theta t))_{jk}y_k$$

where we also used invariance of the labels: $\Pi(b)\mathcal{Y}_B = \mathcal{Y}_B$. Plugging this into the expression above:

$$|\bar{\pi}_t(s) - \bar{\pi}_t(\psi_S(g)s)| = |(\Theta(s, s_{\mathrm{p}_b^{-1}(i)}) - \Theta(\psi_S(g)s, s_i))\Theta_{ij}^{-1}(\mathbb{I} - \exp(-\eta\Theta t))_{jk}y_k|$$
$$= |(\Theta(s, s_{\mathrm{p}_b^{-1}(i)}) - \Theta(s, \psi_S^{-1}(g)s_i))\Theta_{ij}^{-1}(\mathbb{I} - \exp(-\eta\Theta t))_{jk}y_k|$$

Now, we bound the following:

$$\Delta\Theta(s', s, \bar{s}) = |\Theta(s', s) - \Theta(s', \bar{s})|$$
$$= \left| \sum_{l=1}^L \mathbb{E}_{\theta \sim \mu} \left[ \left( \frac{\partial \pi_\theta(s')}{\partial \theta^{(l)}} \right)^\top \left( \frac{\partial \pi_\theta(s)}{\partial \theta^{(l)}} - \frac{\partial \pi_\theta(\bar{s})}{\partial \theta^{(l)}} \right) \right] \right|$$
$$\leq ||s - \bar{s}|| \sum_{l=1}^L \mathbb{E}_{\theta \sim \mu} \left[ \left| \left( \frac{\partial \pi_\theta(s')}{\partial \theta^{(l)}} \right)^\top \cdot L(\theta^{(l)}) \right| \right]$$
$$= ||s - \bar{s}||\hat{C}(s')$$

where $L(\theta^{(l)})$ is the Lipschitz constant of $\partial_{\theta^{(l)}}\pi_\theta$. Finally, using the triangle inequality:

$$|\bar{\pi}_t(s) - \bar{\pi}_t(\psi_S(g)s)| \leq \hat{C}(s)\sqrt{\sum_i ||s_{\mathrm{p}_b^{-1}(i)} - \psi_S(g)s||^2} \sqrt{\sum_i (\sum_{j,k} \Theta_{ij}^{-1}(\mathbb{I} - \exp(-\eta\Theta t))_{jk}y_k)^2}$$
$$\leq \kappa\hat{C}(s)\sqrt{\sum_i ||s_i||^2} \sqrt{\sum_i (\sum_{j,k} \Theta_{ij}^{-1}(\mathbb{I} - \exp(-\eta\Theta t))_{jk}y_k)^2} = \kappa C_\Theta(s)$$

$\square$

### E.3 Lemma B.4 from Gerken and Kessel (2024)

Here we repeat the proof for Lemma B.4 in Gerken and Kessel (2024) in the notation used in this paper.

**Lemma B.4.** *The probability that the deep ensemble $\bar{\pi}_t$ and its Monte-Carlo estimate $\hat{\pi}_t$ differ by more than a given threshold $\delta$ is bounded by*

$$\mathbb{P}\big[|\bar{\pi}_t(s) - \hat{\pi}_t(s)| > \delta\big] \leq \sqrt{\frac{2}{\pi}}\frac{\sigma_s}{\delta}\exp\left(-\frac{\delta^2}{2\sigma_s^2}\right),$$

*where we have defined*

$$\sigma_s^2 := Var(\hat{\pi}_t)(s) = \frac{\Sigma_t(s)}{N}$$

*where $\Sigma_t(s)$ is the NNGP variance and $N$ is the finite ensemble size.*

*Proof.* In the infinite width limit, the ensemble members for our Monte-Carlo estimator $\hat{\pi}_t$ are i.i.d. random variables drawn from a Gaussian distribution with mean $\bar{\pi}_t$ and variance $\Sigma_t$. Therefore, the probability of deviation for a given threshold $\delta$ is given by

$$\mathbb{P}\big[|\bar{\pi}_t(s) - \hat{\pi}_t(s)| > \delta\big] = \frac{2}{\sqrt{2\pi}\sigma_s}\int_\delta^\infty \exp\left(-\frac{x^2}{2\sigma_s^2}\right)\mathrm{d}x\,,$$

where $\sigma_s^2 = \frac{\Sigma_t(s)}{N}$ is the variance of the Monte-Carlo estimator $\hat{\pi}_t$. With a change of integration variable $u = \frac{t}{\sigma_s\sqrt{2}}$, and using the fact that $1 \leq \frac{2u}{2\min(u)}$ for $u \geq \min(u)$, we get:

$$\mathbb{P}\big[|\bar{\pi}_t(s) - \hat{\pi}_t(s)| > \delta\big] = \frac{2}{\sqrt{\pi}}\int_{\frac{\delta}{\sqrt{2}\sigma_s}}^\infty \exp\left(-u^2\right)\mathrm{d}u \leq \frac{1}{\sqrt{\pi}}\frac{\sqrt{2}\sigma_s}{\delta}\int_{\frac{\delta}{\sqrt{2}\sigma_s}}^\infty (2u)\exp\left(-u^2\right)\mathrm{d}u\,.$$

Finally, this integral evaluates to

$$\mathbb{P}\big[|\bar{\pi}_t(s) - \hat{\pi}_t(s)| > \delta\big] \leq \sqrt{\frac{2}{\pi}}\frac{\sigma_s}{\delta}\exp\left(-\frac{\delta^2}{2\sigma_s^2}\right)$$

$\square$

