# OpenReview forum: "How Ensembles of Distilled Policies Improve Generalisation in Reinforcement Learning"
_NeurIPS.cc/2025/Conference — NeurIPS 2025 poster_

### Official Review · Reviewer_3YDA · 2025-07-01

**Clarity:** 3
**Significance:** 3
**Originality:** 3
**Rating:** 4
**Confidence:** 3

**Summary:**

The paper studies policy transfer in RL environments. Assume we have a contextual MDP, where we are allowed to observe training contexts $C_{train}$ but wish to generalize to test contexts $C_{test}$. It has been shown in practice that sometimes, distilling a policy can lead to a policy with better generalization behavior than the original policy. This paper is primarily a theory paper, deriving a generalization bound on distillation, assuming finite and infinite ensembles of policies. Note that the paper views BC as a specific instance of policy distillation (distillation of the optimal policy).

The proof focuses on a specific subcase where the contexts of the contextual MDPs are elements of a symmetry group $G$, and the training contexts are elements of a subgroup $B$. This assumption allows the authors to define a constant $\kappa$, which measures the
discrepancy between subgroup $B$ and the entire group $G$. (Intuitively, a measure for how "close" the training contexts $B$ are to test time contexts in $G - B$.

The paper further assumes an infinitely wide neural network, following the NTK line of work, and an infinite number of training samples from the training contexts $C_{train}$. This allows the authors to prove a bound between the optimal return $J^{\pi^*}$ and return of the ensemble of $N$ policies $J^{\pi_\infty}$, that depends on:

* A constant term that depends on the discount factor, and Lipschitz constants for transitions, rewards, and policies.
* An additive constant based on $\kappa$ and the network architecture, which is irreducible error even if number of ensembled models $N$ approach infinity.
* And one term that scales with $O(1/\sqrt{N})$ on the number of models in the ensemble.

In short, the bound grows tighter as subgroup $B$ covers more of the group $G$, and as number of models in the ensemble increases.

To evaluate how well this appears practically, the authors study a 2D reacher environment, where the symmetry group is the SO(2) group of rotations, and the subgroups are $C_2, C_4, C_8$ corresponding to 180 / 90 / 45 degree rotations around the circle.

The authors finally demonstrate that widening the initialization (context for the MDP) causes better test time generalization, as does changes to the makeup of the trajectory dataset to include more exploratory data.

**Questions:**

Could the authors explain why Distillation (Teacher dataset) performs so differently than Behavior Cloning (Teacher dataset)? Shouldn't these two be identical? As the author state earlier, they consider behavior cloning to be a special case of distillation.

**Ethical Concerns:**

["NO or VERY MINOR ethics concerns only"]

**Final Justification:**

Did not have many issues with the original paper, believe paper is technically solid and authors have addressed my concerns.

**Limitations:**

yes

**Quality:**

3

**Strengths And Weaknesses:**

Most of the results in this work are intuitively understood, but this paper proves more formal bounds on expected generalization of policies trained via distillation. The paper is helpful on providing references to background theoretical material, and although I did not read the proofs very closely, at a high level nothing about them seemed wrong / I expect them to hold if checked more closely.

The subgroup + symmetry group assumption in particular is fairly restrictive however. Given that the purpose of this assumption is solely to define $\kappa$, it's not clear to me that this actually adds much to the proof either? I would assume that instead of requiring a symmetry group, you could instead directly define $\kappa$ as an unobserved constant measuring the distance between $C_{train}$ and $C_{test}$, along with the appropriate Lipschitz assumptions, and get an equivalent theorem that's more general.

I think it is good that the authors show that test performance improves as the ensemble grows in side, even while train-time performance has saturated (Section 5.1). However I was a bit disappointed that there was no attempt made to check how closely this follows the $1/\sqrt{N}$ derived earlier. If you abstract Eqn (5) to $a(b + \frac{c}{\sqrt{N}})$ for unknown constants $a,b,c$ and try to fit lines to it, surely that would be a better argument for how good the bound in (5) is? As it stands there is no intuition for whether it's tight or loose. Similar things could be said about measuring $\kappa$.

Finally I'm not sure Section 5.2.1 does anything for the paper, since as far as I can tell it has little bearing on the theoretical result proven by the paper. Taken on its own as an empirical result, it argues that using a wider distribution of training data leads to better test time performance in environments where we fully saturate the training set, but this is already a well-known result from the rest of deep learning. So it seems irrelevant to the paper itself.

---

> ### Author Rebuttal · Authors · 2025-07-30
>
> We thank the reviewer for their insightful comments and questions, which have helped us to improve the clarity and impact of our paper. We address each point below.
> - **Regarding distillation vs. behaviour cloning:** This indeed raises an important point about the difference between distillation and behaviour cloning as defined in our paper. The key difference lies in their optimisation objectives and learning targets. Both aim to imitate the same behaviour policy, but behaviour cloning can only observe the sampled actions from that policy, whereas distillation has access to all action probabilities. This implies different losses: behaviour cloning uses a cross-entropy loss, whereas distillation regresses the probabilities with a MSE loss (see Appendix A.1.1). As a result, distillation *should* be much more sample efficient, as it has access to more information than behaviour cloning.
> - **Regarding the $\frac{1}{\sqrt{N}}$ generalisation bound:** We appreciate the suggestion to analyse how closely our results align with the $\frac{1}{\sqrt{N}}$ theoretical bound. This is indeed an interesting analysis that could provide valuable intuition. While we believe this would be a valuable addition, conducting the necessary experiments would unfortunately before the current discussion period ends. However, we are committed to performing this analysis and will include it in the final version of the paper. That being said, we do want to manage expectations regarding the tightness of this bound. Obtaining tight generalisation bounds for neural networks is notoriously challenging [1,2]. Furthermore, some of our theoretical assumptions, such as infinitely wide neural networks, are hard to meet in practice. We believe the true strength of our theory lies in its ability to identify crucial properties of the dataset distribution and distilled ensemble that are capable of improving generalisation performance. Moreover, our paper introduces a novel analytical approach to policy distillation, paving the way for more advanced theoretical developments in the future.
> - **Regarding section 5.2.1:**  While it is true that these experiments break a significant number of the theoretical assumptions, we firmly believe this section is relevant to the paper's overall message. Section 5.2.1 serves to empirically demonstrate how our insights can be leveraged to significantly enhance the generalisation performance of a reinforcement learning agent through policy distillation. A clear example of this is seen in our ensemble $N=10$, distilled on the Explore-Go dataset, which achieves substantially higher test performance than the original PPO+Explore-Go teacher policy (see Table 3). The potential of policy distillation to improve generalisation was initially identified in [3], but we believe our paper provides a more compelling argument and empirical evidence for this phenomenon. Thus, rather than aiming to confirm theoretical predictions, Section 5.2.1 showcases a significant empirical finding for which our theory provides the most convincing justification and motivation to date. Furthermore, the idea of broadening the training distribution by sampling more extensively from existing training environments, as opposed to generating entirely new ones, is an emerging concept in the RL literature, recently introduced in [4,5]. Our paper demonstrates that this concept also applies effectively to policy distillation, and to the best of our knowledge, we are the first to highlight these specific generalisation benefits within this context.
>
> To improve the clarity of the paper, we will add a dedicated limitations section that will discuss all the limitations and points raised during review. We hope the clarifications outlined here adequately address the reviewer's concerns and demonstrate the potential of our approach. If not, we are of course open to discuss any further questions during this discussion phase.
>
> **References** \
> [1] Fantastic Generalization Measures and Where to Find Them. Jiang et al. 2020 \
> [2] Fantastic Generalization Measures are Nowhere to be Found. Gastpar et al. 2023 \
> [3] Learning Dynamics and Generalization in Reinforcement Learning. Lyle et al. 2022 \
> [4] On the Importance of Exploration for Generalization in Reinforcement Learning. Jiang et al. 2023 \
> [5] Exploration Implies Data Augmentation: Reachability and Generalisation in Contextual MDPs. Weltevrede et al. 2025

---

> > ### Comment · Reviewer_3YDA · 2025-08-07
> >
> > Thanks for the comments on the paper. I did forget that the generalization bound assumes an infinitely wide neural net, which would be hard to actually do experiments with. I plan to keep my score the same.

---

### Official Review · Reviewer_QfQd · 2025-07-01

**Clarity:** 3
**Significance:** 2
**Originality:** 3
**Rating:** 4
**Confidence:** 4

**Summary:**

This paper explores the generalization ability of policy distililation both theoretically and empirically, showing that ensemble of policy distilation on augmented data improves generalization.

**Questions:**

1.Is the ensemble used during training or during testing? Which contributes more?

2.Could the gains stem primarily from regularization (e.g., via ensemble averaging or noise injection) rather than group invariance? How do experiments isolate invariance as the key factor?

3. Is there a principled way to determine the minimal sufficient diversity/size of the distillation dataset for a task? Does the theory offer guidelines beyond "more is better"?

**Ethical Concerns:**

["NO or VERY MINOR ethics concerns only"]

**Final Justification:**

I will maintain my original rating and recommend accept. All my concerns have been addressed.

**Limitations:**

yes

**Quality:**

2

**Strengths And Weaknesses:**

**Strengths**
1. The paper proposes a novel theoretical framework (GTI-ZSPT) that formalizes how group invariance and ensemble distillation contribute to policy generalization.
2. Theoretical results are non-trivial and connect deeply with recent findings in deep learning theory and group invariance.
3. The exposition is clear, with thoughtful examples and visual aids.

**Weaknesses**
1. There is a large gap between theory and practice: the theory focuses on the size of the *symmetry group* of the dataset, but in practice, collecting diverse states will simply benefit, as suggested in Section 5.
2. Using 10-policy ensembles incurs significant training/inference overhead. No analysis is provided for smaller ensembles (e.g., 3–5 policies) or efficiency trade-offs, limiting practical adoption.

---

> ### Author Rebuttal · Authors · 2025-07-30
>
> We appreciate the reviewer's insightful questions, which help to clarify important aspects of our work. Please find our responses to each point below.
> - **Regarding how and when the ensemble is used:** The ensemble members are independently distilled on a single distillation dataset and the forward pass through each ensemble member is performed independently (and the output is then averaged) at each step during testing. Therefore, the ensemble is not used during training; its role is solely during testing to average over the ensemble members' outputs at each step. This approach allows for high parallelisability during both training and inference, keeping computational overhead relatively small. If parallel computation is not possible, the computational overhead would scale linearly with ensemble size. We also note that since all ensemble members are distilled on the same dataset, the number of additional environment interactions required to create this dataset (which is typically considered the main bottleneck in RL) is independent of the ensemble size.
> - **Regarding regularisation versus group invariance:** This is indeed a very important and actively debated point in the data augmentation literature. There is ongoing academic discussion regarding how data augmentation precisely improves generalisation. Some studies suggest that data augmentation, even when performed with respect to a clearly defined group invariance, does not guarantee the learning of an invariant function [1]. Instead, some work suggests that the real underlying mechanism has more to do with reducing overfitting to spurious correlations, than inducing symmetry invariance in the model [2]. As such, whether the benefits of performing data augmentation with respect to some symmetry group actually stem from induced invariance or reduced overfitting and other forms of regularisation, is still an open topic of discussion. Nonetheless, the empirical benefit of data augmentation in various domains and settings is widely acknowledged [3-6], and we demonstrate in Section 5.2.1 that the insights from our theory translate to environments in which the group symmetries are no longer obviously present.
> - **Regarding how experiments isolate invariance as the key factor:** As discussed above, whether the generalisation benefits from data augmentation stem from learned invariance or other mechanisms (e.g., reduced overfitting and regularisation), is still an open topic of research in the data augmentation literature. However, there are attempts made to try to analyse the induced invariance of the model. If the symmetries are known, it is possible to measure the variance of the network output across the group orbit [7]: if the model is invariant, this variance should be zero. Unfortunately, we cannot perform this analysis for the Four Room experiment in Section 5.2, since we do not know the exact symmetries that govern this environment. However, we are committed to perform this analysis for the rotational Reacher experiments in Section 5.1 and add it to the paper. This will not provide a definitive answer, but can give an indication of the level of invariance induced in the model for the Reacher environment that most closely aligns with our theory.
> - **Regarding a principled way to determine the minimal sufficient diversity/size of the distillation dataset for a task:** Following our theoretical framework, the relevant metric to minimise would be the discrepancy measure $\kappa$. For any fixed subgroup size $|B|$, this measure is minimised if the subgroup has maximal coverage over the full group $G$ (as measured by the operator norm in representation space $\psi_S$ ). This aligns with some prior empirical work on data augmentation, where distributing augmentations uniformly leads to good performance [8,9]. However, we are careful not to overstate the predictive power of the current theory, as it relies on assumptions that are difficult to hold in practice (such as the infinitely wide neural networks assumption). We believe the primary strength of our theory lies in its ability to help identify relevant properties of the data distribution and the distillation ensemble that can be leveraged to improve generalisation performance. It serves as a valuable first step that could potentially facilitate further theoretical developments in future work, allowing these types of questions to be answered with greater certainty.
>
> **References** \
> [1] On the Benefits of Invariance in Neural Networks. Lyle et al. 2020 \
> [2] Data Augmentation as Feature Manipulation. Shen et al. 2022 \
> [3]  A survey on image data augmentation for deep learning. Shorten and Khoshgoftaar 2019 \
> [4] A survey of data augmentation approaches for NLP. Feng et al. 2021 \
> [5] Generalization in Reinforcement Learning by Soft Data Augmentation. Hansen et al. 2021 \
> [6] Automatic data augmentation for generalization in reinforcement learning. Raileanu et al. 2021 \
> [7] In What Ways Are Deep Neural Networks Invariant and How Should We Measure This? Kvinge et al. 2022 \
> [8] UniformAugment: A Search-free Probabilistic Data Augmentation Approach. Lozhkov et al. 2020 \
> [9] TrivialAugment: Tuning-free Yet State-of-the-Art Data Augmentation. Müller and Hutter 2021

---

> > ### Comment · Reviewer_QfQd · 2025-08-05
> >
> > Thank you for your thorough rebuttal. Your response clarifies most of my concerns and demonstrates that this paper presents an important theoretical result. However, it does not provide sufficient novel insights regarding the application of this theoretical result to practical algorithms; this limitation reduces the broader impact of the work. Therefore, I will maintain my original score as boarderline accept.

---

### Official Review · Reviewer_q7PZ · 2025-07-03

**Clarity:** 3
**Significance:** 3
**Originality:** 3
**Rating:** 5
**Confidence:** 3

**Summary:**

This paper aims to investigate, under a zero-shot policy transfer setting (ZSPT) in reinforcement learning (RL), why policy distillation leads to more generalizable policies and what data should be used for distilling the policy. Specifically, the authors prove a generalization bound for a distilled policy in a generalization through invariance ZSPT (GTI-ZSPT) setting.  The bound provides two insights:
1. If the policy distillation is performed on an ensemble of N infinitely wide neural networks, a larger N leads to a lower generalization bound.
2. The generalization bound can be lowered by distilling the policy with more data from the environments.

The experiments validate that increasing the number of ensemble policies or training on a larger subgroup yields better results. In addition, the experiments also demonstrate that some of the assumptions can be relieved while the insights provided by the theory still hold.

**Questions:**

My questions are mostly mentioned in the weaknesses/suggestions section.
1. What types of real-world data satisfy or do not satisfy the symmetric property?
2. How to choose the smallest subgroup that can achieve good enough generalizability?
3. What is the relationship between the number of ensemble policies and performance improvement? Is it linear or non-linear?
4. Do you expect the insights provided by your theory to hold in more complex environments, e.g., Playground (in Minigrid) or robotics tasks?

**Ethical Concerns:**

["NO or VERY MINOR ethics concerns only"]

**Final Justification:**

I have read through all the reviews and authors’ responses. After careful consideration, I decided to raise my rating accordingly, as the proposed theoretical findings are crucial to gaining a deeper understanding of how commonly applied techniques, e.g., data augmentation, work or can be improved.

**Limitations:**

It is difficult to tell which part of Section 5 mentions limitations. Please try to have a separate section discussing the limitations.

**Quality:**

3

**Strengths And Weaknesses:**

# Strengths
This paper is theoretically sound in its discussion of the key factors that facilitate generalizable policy transfer in RL through policy distillation. It provides mathematical validation of the effects of ensemble policies and the diversity of the data trained on, which are intuitive results, but opens the door to more advanced algorithm development.

In the experiments, the paper also demonstrates that even if the assumptions required by the theory do not hold, we can still observe the effects caused by the number of ensemble policies and diverse training data. This improves the applicability of the theoretical results.

# Weaknesses/Suggestions
1. The theory primarily relies on the symmetries present in many real-world data distributions. However, it is not straightforward to determine which types of real-world data satisfy this assumption. In addition to the Lie group SO(2) example provided in the paper, there can be more discussion on other data types that have symmetric properties.
2. The theory suggests that if the training data covers a larger subgroup, the generalizability will be improved. I understand that in practice, it is challenging to train the policy on the entire group G; therefore, the experiments also suggest that random data can enhance generalizability. Yet, based on the theoretical results, I am interested in seeing how to wisely choose a subgroup that can achieve good enough generalizability. This discussion brings insights into balancing sample efficiency and performance. I am also interested in learning how to determine the number and diversity of ensemble policies.
3. There can be more experiments on environments that do not have invariance symmetry, as validating in one environment might not be sufficient to validate your conclusions.

---

> ### Author Rebuttal · Authors · 2025-07-30
>
> We sincerely thank the reviewer for their thoughtful comments and insightful questions, which have helped us clarify and strengthen our manuscript. We've carefully considered each point and provide the following responses.
> - **Regarding what types of real-world data satisfy or do not satisfy the symmetric property:** The real world is rich with symmetries, including mirror, translational, and scaling symmetries, or simply invariances to noise or irrelevant background details. For example:
> 	- **Image recognition** heavily relies on these symmetries, where objects are identifiable regardless of their orientation, position or scale.
> 	- **Robotics** often deals with actions and environments exhibiting symmetries in physics or optimal paths. For example, rotational and translational symmetries when moving from A to B.
>
> Additionally, there might be many more symmetries existing in the real world that we are not immediately aware off since they are not obvious from our current viewpoint. We believe the strength of our approach lies in identifying insights that can hold even when the symmetries are no longer obvious to us, as demonstrated in the Four Rooms experiments in Table 3.
> - **Regarding the smallest subgroup that can achieve good enough generalisability:** Our theory suggests minimising the discrepancy measure $\kappa$. For any fixed subgroup size $|B|$, this measure is minimised when the subgroup has maximal coverage over the full group $G$ (as measured by the operator norm in representation space $\psi_S$). This aligns with some prior empirical work on data augmentation, where distributing augmentations uniformly leads to good performance [1,2].
> - **Regarding the relationship between the number of ensemble policies and performance improvement:** Our theory indicates that the performance of a distilled ensemble is bounded by optimal performance with a term of the form $b + \frac{c}{\sqrt{N}}$, where $N$ is the ensemble size. This shows a non-linear relationship, with diminishing returns as $N$ increases. This relationship provides an upper bound on the performance difference, so it might not perfectly predict actual realised performance improvement. However, we are careful not to overstate the predictive power of the current theory, as it relies on simplifying assumptions (like infinitely wide neural networks) that are hard to meet in practice. We believe the true power of our theory lies in helping identify relevant dataset properties and distillation ensemble characteristics for improved generalisation, rather than providing exact quantitative predictions for ensemble size versus performance. However, we believe our work is an important first step that could lead to more precise theoretical and empirical developments in future work.
> - **Regarding more complex environments:** While our theory focuses on specific symmetric structures, we believe its insights could hold much more generally in complex environments like Minigrid's Playground or robotics tasks. For example, our rotational Reacher experiment (Table 2 in Section 5.1.1) saw no significant difference between training on an exact subgroup $B=C_4$ (as per theory) versus random augmentations. From our understanding, this aligns with the known gap between theoretical and empirical benefits of data augmentation. Often theoretical results require global group symmetry [3,4], yet empirical evidence shows data augmentation improves generalisation even when augmentations aren't strictly symmetric [5-7] or aren't global [8,9]. Therefore, we believe our insights have the potential to boost generalisation in even more complex settings. Further analysis of what environmental properties are required for our insights to hold is a very interesting direction for future work.
>
> Overall, we agree more discussion on the limitations of our work would strengthen the paper, and we will add a dedicated limitations section to the to accomplish this. We hope these clarifications adequately address the reviewer's concerns and demonstrate the potential of our approach. We are open to discuss any further questions during the discussion phase.
>
> **References** \
> [1] UniformAugment: A Search-free Probabilistic Data Augmentation Approach. Lozhkov et al. 2020 \
> [2] TrivialAugment: Tuning-free Yet State-of-the-Art Data Augmentation. Müller and Hutter 2021 \
> [3] On the Benefits of Invariance in Neural Networks. Lyle et al. 2020 \
> [4] A Group-Theoretic Framework For Data Augmentation. Chen et al. 2020 \
> [5] On the Generalization Effects of Linear Transformations in Data Augmentation. Wu et al. 2020 \
> [6] Generalization in Reinforcement Learning by Soft Data Augmentation. Hansen et al. 2021 \
> [7] Automatic data augmentation for generalization in reinforcement learning. Raileanu et al. 2021 \
> [8] Data Augmentation as Feature Manipulation. Shen et al. 2022 \
> [9] Learning Instance-Specific Augmentations by Capturing Local Invariances. Miao et al. 2023

---

> > ### Comment · Reviewer_q7PZ · 2025-08-05
> >
> > Thank you for your responses. The additional explanations have provided interesting insights for this work.
> >
> > I have read through all the reviews and authors’ responses. After careful consideration, I decided to raise my rating accordingly, as the proposed theoretical findings are crucial to gaining a deeper understanding of how commonly applied techniques, e.g., data augmentation, work or can be improved.

---

### Official Review · Reviewer_3N2K · 2025-07-03

**Clarity:** 3
**Significance:** 3
**Originality:** 3
**Rating:** 5
**Confidence:** 4

**Summary:**

The paper provides a group-theoretical understanding of policy distillation in RL when training a policy on a diverse set of contexts. It is shown that it is better to train an ensemble of distilled policies rather than a single policy. Further, when the data distribution exhibits symmetries, invariances introduced into the training data distribution tighten a generalisation bound on the distilled policy performance.  A reacher task with rotational invariances and a four-room navigation problem are provided as benchmarks, showing that larger ensembles and larger set of invariances boost performance.

**Questions:**

1. How does the runtime increase in the size of the ensemble?

2. For more complex tasks that do not possess obvious invariances as the reacher experiment, how should the subgroup B be selected? Can this be learned from the given data?

3. How does the method perform on large-dimensional problems?

**Ethical Concerns:**

["NO or VERY MINOR ethics concerns only"]

**Final Justification:**

I am satisfied with the rebuttals which answered all the main concerns of all the reviewers. Overall I really enjoyed reviewing this paper, and I believe the theoretical insights into how and why data augmentation works for imitation and policy distillation will be important in moving RL forward. I will therefore maintain my positive score.

**Limitations:**

I think more discussion should be provided about possible failure modes of the method in real-world settings where good invariances might be difficult to determine. I think a more detailed discussion of runtime costs, especially pertaining to ensemble size and how to mitigate it, could be useful.

**Quality:**

4

**Strengths And Weaknesses:**

Strengths:

1. The generalisation bound for distilled ensembles under subgroup/symmetry coverage is novel and mathematically elegant.

2. The paper provides clear takeways from the theory that can be translated into practice: distill into an ensemble and use diverse states. While seemingly simple concepts, the theoretical understanding of why these things matter for policy distillation performance are less trivial.

3. Ablations on ensemble size and distillation-data diversity convincingly support the theory.

Weaknesses and comments:

1. It seems that runtime should also increase significantly as a function of N, the size of the ensemble. It would be good to also show the runtimes associated with N = 1, N = 10 and N = 100 in Section 5.1.

2. While the theory suggests that larger subgroups result in a lower performance gap, based on my understanding, the current experiments show it suffices to generate maximally diverse data from many invariances to improve performance. However, for more complex tasks that do not possess such obvious invariances as the reacher experiment, it remains largely unclear to me how the subgroup B should be constructed in practice. Do you suggest to do data augmentation with random invariances, hoping to generate data that sufficiently covers the testing distribution? What happens if you apply the wrong invariance for the problem or an invariance that fails to adequately capture the variation of the test instances? Does the method fail? Are there more principled ways to extract a (small) set of invariances that provide maximum coverage/generalization?  I think a section discussing the limitations and strategies to address these limitations would be helpful. Ideally, an experiment showing sensitivity of the method to poorly chosen invariances would be even more insightful so we can better understand what to focus on in the future.

3. The existing experiments are somewhat limiting and focused on 2-D problems. I think some additional experiments on higher dimensional problems (i.e. Mujoco/D4RL) where invariances may be less trivial to design would be helpful.

---

> ### Author Rebuttal · Authors · 2025-07-30
>
> We thank the reviewer for their insightful comments and questions, which have helped us to further clarify and strengthen our manuscript. We have addressed each point below.
>
> - **Regarding the runtime increase of the ensemble:** All ensemble members are designed to be trained independently, and during inference, they can also be evaluated independently (an independent forward pass with an average over the output of the ensemble afterwards). This inherent independence means that both the training and inference processes of the ensemble are fully parallelisable. If parallelisation is not feasible, the runtime for both training and inference would increase linearly with the ensemble size. Furthermore, it is important to note that all policies are distilled using the same distillation dataset. This means the small number of additional environment steps required to sample a more diverse dataset is independent of ensemble size.
> - **Regarding more complex tasks that do not possess obvious invariances:** This is an important question, and we agree that for more complex tasks, learning the invariant subgroup B from the data could be a potential solution. However, at this stage, we are not entirely convinced that this approach will offer significant benefits. In our experiments with the Reacher environment, we observed no substantial advantage from training on a specific subgroup B compared to simply training on random transformations (see Table 2). While the Reacher environment is relatively small and conclusions should not be overly generalised, this finding aligns with our understanding of how data augmentation operates in practice. Although performing data augmentation with group symmetrical assumptions is necessary for certain theoretical guarantees [1,2], there seems to be a gap between the theoretically proven and the empirically observed benefits of data augmentation. In fact, a large body of empirical evidence suggests that data augmentation improves generalisation outside of such strict assumptions on the structure of the augmentations and underlying data distribution. For example, generalisation benefits are observed _even when_ the augmentations result in distorted data outside the support of the true data distribution (e.g., [3-6]). As such, we believe that collecting a more diverse dataset for distillation has the potential to increase generalisation performance, even if the additional data does not perfectly align with the testing distribution. However, we acknowledge that a more principled approach to creating such a diverse distillation dataset could more efficiently lead to increased generalisation in more complex domains (beyond our experiment in the rotational Reacher environment). To accomplish this, we believe the further analysis of the existing and future literature on the theoretical benefits of data augmentation is an exciting direction for future work.
> - **Regarding performance on large-dimensional problems:** This is an excellent question that points to a critical area for future research. It is indeed still an open question how well our method performs on more complex, higher-dimensional problems where the invariance may be less obvious. We believe our experiments on the Four Rooms environment (Table 3) represent a crucial first step in addressing this. In this environment, the exact symmetries governing the data distribution are not immediately obvious to us, yet we still observed significant benefits from distilling an ensemble ($N=10$) on more diverse data. This was evident with two distinct approaches to attaining this diversity: the Explore-Go dataset and the Mixed dataset. We are confident that further analysis into what properties of environments and distillation datasets could lead to improved performance will be a very interesting and fruitful direction for future work.
>
> Overall, we agree more discussion on the limitations of our work would strengthen the paper even more, and we will add a dedicated limitations section to achieve this. We hope these clarifications adequately address the reviewer's concerns and demonstrate the potential and significance of our approach. We are open to discuss any further questions during the discussion phase.
>
> **References** \
> [1] On the Benefits of Invariance in Neural Networks. Lyle et al. 2020 \
> [2] A Group-Theoretic Framework For Data Augmentation. Chen et al. 2020 \
> [3] The good, the bad and the ugly sides of data augmentation: An implicit spectral regularization perspective. Lin et al. 2024 \
> [4] How Much Data Are Augmentations Worth? An Investigation Into Scaling Laws, Invariance, and Implicit Regularization. Geiping et al. 2023 \
> [5] On the Effects of Artificial Data Modification. Marcu and Prugel-Bennet 2022 \
> [6] On the Generalization Effects of Linear Transformations in Data Augmentation. Wu et al. 2020

---

> > ### Comment · Reviewer_3N2K · 2025-08-05
> > **Thank you for clarifications**
> >
> > I'd like to thank the Authors for a very interesting and comprehensive rebuttal. Overall, this is one of the most interesting papers on data augmentation and general RL I have read this year. I think future work should focus on designing more careful benchmark data sets/problems for evaluating data augmentation in the RL space, as it should not be the main focus of this paper. I am convinced by the responses to my questions and those of the other reviewers, and I will therefore keep my positive score.

---

### Decision · Program_Chairs · 2025-09-17

**Decision:**

Accept (poster)

**Comment:**

(a) Scientific Claims and Findings

The central claim of this paper is that the generalization of a distilled policy in RL can be provably improved by using an ensemble of student policies. The authors theoretically “analyse the act of distilling a policy after training, and try to answer 35 how the policy should be distilled and on what data”. They formalise this by proving a generalization bound in a "Generalisation Through Invariance" setting, where the performance gap to an optimal policy is shown to decrease with the ensemble size (N) and with better coverage of the underlying symmetry group (a smaller discrepancy measure, k). The empirical findings confirm these two principles hold in practice, both in a setting that matches the theory and in a more general setting that violates the theory's strict assumptions.

(b) Strengths

The paper's primary strength is its novel theoretical contribution as it provides a formal answer to the important question of why 1) policy distillation can improve generalisation, moving beyond prior empirical observations and indirect explanations, 2) data augmentation via invariances. The resulting insights—use ensembles and diverse data—are simple, intuitive, and directly actionable. I found the connection made between generalization in RL, group theory, and the neural tangent kernel (NTK) literature both elegant and insightful. The experiments are well-designed to specifically test the predictions of the theory and demonstrate their robustness when assumptions are relaxed.

(c) Weaknesses

The main weakness of the paper lies in the strong assumptions required to prove the theoretical result, including Lipschitz continuity of the MDP, the existence of a clear symmetry group, and the use of infinitely wide neural networks. This creates a noticeable gap between formal theory and messy real-world applications. The small experiments demonstrate empirically that the qualitative insights from the theory are still predictive and useful in a more practical setting (Minigrid's Four Rooms) where these assumptions do not hold, however the method was not tested in more challenging, less artificial settings.

(d) Reasons for Decision

The paper is recommended for acceptance because the paper is technically solid and offers clear takeaways that are backed by well-executed experiments. It also brings together various results in DL and empirical observation, thus it advances our understanding of why certain techniques work, which is a valuable contribution to the field. The reviewers were unanimously positive, and their initial concerns were effectively addressed during the rebuttal period, leading to a strong consensus.

(e) Discussion and Rebuttal

The author-reviewer discussion was highly constructive. Authors addressed all major points, clarifying distinctions and agreeing to add an empirical analysis. Their effective engagement strengthened the paper and resolved reviewer doubts.